# Exploring ambivalence: A psychoanalytic analysis of emotional complexity in selected Amharic novels

Dawit Dibekulu[1]*, Tesfaye Dagnew[1], Tesfamaryam G/Meskel[2]

**1** Department of English Language and Literature, Faculty of Humanities, Bahir Dar University, Bahir Dar, Ethiopia, **2** Department of English Language and Literature, Faculty of Humanities, Literature, Bahir Dar University, Bahir Dar, Ethiopia

* dawitdibekulu7@gmail.com

## Abstract

This study examines the representation of ambivalence in the Amharic literary canon, focusing specifically on the novels Yäqënat Zār (Zar of Jealousy) and Emba ɨnna Saq (Tears and Laughter). While previous research has explored various themes within Amharic literature, there is a notable gap in the exploration of ambivalence as a central emotional and psychological force in these narratives. This research seeks to address this gap by analyzing the multifaceted expressions of ambivalence and its consequences within the selected novels. Through a psychoanalytic lens, it delves into how personal loss, cultural anxieties, and the complexities of identity intersect to shape the characters' experiences. The study's use of a qualitative approach within an interpretive paradigm is significant because it allows for an in-depth exploration of the meanings and implications embedded in the text. Close reading, as the primary method, enables a detailed analysis of the text's language, structure, and themes, helping to uncover layers of meaning that extend beyond surface-level understanding. The descriptive and analytical research designs work together to identify key features of the text and analyze how these elements interact to convey deeper insights into cultural, historical, and social contexts. Ultimately, this approach is essential for interpreting how literature reflects human experiences and societal values. Close textual analysis reveals how ambivalence manifests in the characters' internal conflicts, regrets, and anxieties, shedding light on the complexities of their psyches. The findings highlight ambivalence as a driving force that reflects the tensions between cultural expectations, societal pressures, and individual desires. This research underscores the importance of ambivalence in Amharic literature, offering a richer understanding of the emotional and psychological depth within these narratives. The study also suggests that further exploration of ambivalence in other genres of Amharic literature is warranted, taking into consideration the unique cultural and national contexts that shape these emotional experiences.

**Data availability statement:** The data is available with in the article.

**Funding:** The author(s) received no specific funding for this work.

**Competing interests:** The authors have declared that no competing interests exist.

## 1. Introduction

Literature, as a product of human thought, has long been considered a mirror of human life, reflecting the complexities of culture, beliefs, and traditions. As Meiliana [1] suggests, literary works across various genres, such as theater, poetry, prose, novels, and short stories, offer a depiction of human experiences, emotions, and social realities. Literature is not just a form of artistic expression; it also functions as a vital record of human history, reflecting individual and collective struggles, desires, and aspirations. This interdisciplinary nature of literature allows it to intersect with various academic fields, including history, philosophy, sociology, psychology, and even the fine arts. As Meiliana [1] and Russell [2] note, literary studies often transcend national, temporal, and linguistic boundaries, bringing together diverse perspectives to understand human experience across time and space.

One of the most significant intersections within literary studies is between literature and psychology, both of which are concerned with human beings, their emotions, perceptions, and reactions to the world around them [3]. Psychology plays a pivotal role in the analysis of literary texts, especially through the lens of psychoanalytic theory. According to Meiliana [1], Freudian psychoanalysis offers valuable insights into the unconscious mind, revealing not only the personal psychology of the author but also the underlying emotional and psychological states embedded within the text itself. Literary works are often seen as manifestations of the author's unconscious desires, fears, and traumas, and the psychoanalytic critic examines these works as if they were dreams, seeking to uncover the repressed elements of the author's psyche [4]. In this sense, literature becomes a psychological artifact, offering both an intimate glimpse into the author's emotional world and a broader reflection of the human condition.

This research aims to address this gap by exploring the role of ambivalence, mourning, and melancholy in two Amharic novels: Sisay Nigusu's *Yäqënat Zar* [5] and Gebyehu Ayele's *Emba ɨnna Saq* [6]. These novels provide rich textual landscapes for investigating how these psychological and emotional themes are intricately woven into the social, cultural, and personal dimensions of Ethiopian life. Ethiopia's history, marked by periods of political upheaval, social transformation, and economic struggle, has deeply influenced the emotional landscape of its people. In this context, mourning and melancholy are not just personal psychological states but are intertwined with collective experiences of loss, trauma, and societal change. Through the lens of psychoanalytic literary theory, this study seeks to uncover how these novels portray the complex emotional lives of their characters, particularly in their interpersonal relationships. The social fabric of Ethiopia, shaped by its unique blend of traditional values, religious beliefs, and contemporary challenges, forms the backdrop against which these emotional states are experienced. Themes of mourning often echo societal mourning for lost political stability or cultural identity, while melancholia reflects the inability to reconcile individual desires with societal expectations or historical injustices. Ambivalence, in this context, becomes a crucial element that reflects the tension between personal aspirations and collective realities, as well as the struggle to make sense of a rapidly changing world. By

examining how mourning, melancholy, and ambivalence manifest in the characters' emotional and psychological development, the research will offer a deeper understanding of the human condition in the Ethiopian context. It will explore how these themes are not merely individual psychological experiences but also responses to larger socio-political and cultural dynamics. In doing so, this study contributes to a broader understanding of how literature can reflect the complexities of identity, trauma, and emotional resilience within Ethiopian society.

The exploration of ambivalence in Ethiopian literature is significant not only because it highlights the psychological complexity of personal relationships but also because it provides new insights into how emotional states are culturally expressed and understood in Ethiopia. As Tewodros [4] and Ayele [3] have noted, Ethiopian literature, particularly in Amharic, serves as a vital cultural resource, capturing the nation's historical, social, and emotional experiences. In Ethiopia, emotional expression is deeply influenced by a variety of cultural frameworks, including the traditional values of community, family, religion, and the historical legacies of war, revolution, and political instability. How mourning, melancholy, and ambivalence are represented in Ethiopian literature are not merely individual psychological experiences; they are intricately tied to the collective emotional responses to Ethiopia's past and present. For example, mourning in the Ethiopian context may reflect not only personal loss but also collective grief over historical events such as the different regimes' violence, the loss of cultural heritage, or the disruption caused by political and social upheavals. Similarly, melancholia in Ethiopian literature may not only signify personal sorrow but also symbolize an existential struggle to come to terms with Ethiopia's historical transitions and the emotional scars they have left on the people. Thus, this research aims to enhance the understanding of emotional complexity and psychological nuance in Ethiopian literary traditions by exploring the interplay of ambivalence in Amharic novels.

## 2. Statement of the problem

Despite the importance of these themes in literature, there remains a significant gap in the study of mourning, melancholy, and ambivalence within Ethiopian literature, especially in the context of Amharic novels. Globally, various scholars have explored the psychological underpinnings of these emotions within literature. For instance, Schlumpf [7] studied how national trauma is depicted through themes like melancholy and ambivalence in French and Chinese literature and film, while Widyaningrum [8] examined ambivalence in the search for national and postcolonial identity in three novels, applying theoretical frameworks by Benedict Anderson, Homi Bhabha, and Gayatri Spivak. Cosar [9] explored ambivalence in Victorian women's writing, showing how conflicting emotions of love and hate were central to their literary expression. In the same vein, Ula [10] applied Freudian psychoanalysis to *The Kite Runner*, demonstrating how ambivalence in the protagonist's relationship with his close friend was a reflection of the emotional complexities and psychological struggles within the narrative.

In the Ethiopian context, while studies like Kalkidan [11] and Amanuel [12] apply psychoanalytic theory to global and local works, respectively, they do not focus specifically on mourning, melancholy, and ambivalence as central themes within Amharic novels. For example, Kalkidan [11] compared life and death in the poetry of Emily Dickinson and Sylvia Plath, highlighting themes of hopelessness, sorrow, and death. Amanuel [12] applied psychoanalytic theory to Dawit Wendmagegn's *Alemenor*, analyzing the influence of unconscious motives and past experiences on character behavior. Additionally, Mitikie et al. [13] studied post-traumatic stress disorder (PTSD) and emotional healing mechanisms in *Self-Prisoner*, while Mitikie and Tesfaye [14] examined collective trauma and PTSD in *Unresolved Tears*, but both studies primarily focus on trauma and healing, rather than the nuanced emotional dynamics of mourning, melancholy, and ambivalence.

This research seeks to address a gap in Ethiopian literary studies by examining how mourning, melancholy, and ambivalence are portrayed in Amharic novels. These psychological themes are particularly significant in the context of Ethiopian society, given the nation's tumultuous history of conflict, civil war, and ongoing political upheaval. Over the years, Ethiopia has experienced various periods of political instability, including the aftermath of authoritarian regimes, regional conflicts,

and the emotional and social consequences of these events. These ongoing struggles have contributed to a collective emotional landscape marked by grief, loss, and a pervasive sense of uncertainty or ambivalence about the future. The nation's prolonged challenges with political disruption and social change have left deep emotional scars, which are often reflected in the personal and interpersonal dynamics within Ethiopian literature. Amharic novels, as representations of the social, political, and emotional realities of Ethiopian life, provide a unique lens through which these themes can be explored. By focusing on how mourning, melancholy, and ambivalence manifest in personal relationships such as those between parents and children, romantic partners, and friends, this research will examine how these emotional states shape both character development and narrative structure. The novels, by depicting the impact of societal upheavals on individual lives, highlight how complex emotions are intricately linked to the psychological experiences of characters. This research seeks to demonstrate how Ethiopia's political and social changes have shaped these emotional complexities, particularly focusing on the psychological nuances of ambivalence in literature. These psychological states are not merely abstract or universal but are deeply embedded in the specific cultural, historical, and political realities of Ethiopia.

While global scholarship has extensively explored the theme of ambivalence, applying these concepts to Amharic literature offers new insights into how Ethiopian authors depict complex emotions. This research focuses on understanding how these emotional dynamics are portrayed through the psychological experiences of characters, framed within Ethiopia's unique historical and cultural context. Ethiopia's long history of social turmoil, political instability, and trauma has deeply shaped the collective psyche, influencing how emotions like grief, loss, and psychological conflict are expressed [15]. The ongoing social and political challenges, ranging from decades of authoritarian rule to ethnic conflicts, have left an indelible mark on the emotional and psychological landscape of the nation. Amharic novels, in particular, serve as powerful mediums for articulating personal and collective experiences of loss and emotional turbulence. By exploring themes of mourning, melancholy, and ambivalence within the context of personal relationships, whether familial bonds, romantic partnerships, or social connections, these works offer critical insights into how Ethiopians navigate complex emotions in a society marked by continuous upheaval. This research seeks to show how these literary representations are not just reflective of individual psychological states but also deeply tied to the broader, shared historical and cultural experiences of the Ethiopian people [15].

This study is driven by several factors. First, there is a noticeable gap in Ethiopian literary studies regarding the portrayal of mourning, melancholy, and ambivalence, especially with personal relationships. To the best of the researcher's knowledge, no previous study has comprehensively analyzed how these psychological themes are depicted in Amharic literature, particularly in the context of emotional dynamics between individuals. Second, while there is a rich body of international research on these themes in literature, the application of psychoanalytic theory to Amharic novels, especially in the context of personal relationships, remains underexplored. Third, the intellectual curiosity to explore the psychological subtleties of emotions often concealed beneath societal norms and expectations is a key motivator for this study. The intricate emotional landscapes of dyadic relationships, where love, loss, and ambivalence intertwine, offer a valuable avenue for exploring the psychological and emotional complexities that shape human behavior.

Moreover, Ethiopia's history, shaped by conflict, famine, and revolution, provides a poignant backdrop for examining ambivalence. These experiences resonate deeply with Ethiopians, particularly in the aftermath of civil wars and political instability. Amharic novels, reflecting these lived realities, offer valuable psychological insights into how ambivalent emotions are experienced, expressed, and internalized within individuals. This study aims to uncover how these psychological complexities are portrayed in Amharic novels, focusing on their impact on personal relationships and character development within Ethiopia's cultural context.

Lastly, the study's theoretical foundation is based on psychoanalytic literary criticism, which provides useful instruments for examining texts' unconscious, suppressed emotions, and psychological dynamics. Finding the hidden layers of meaning in literature, particularly with regard to complicated emotions and ambivalence, has been made possible in large part by psychoanalytic approaches. This study will advance our knowledge of how these emotions act within the larger

narrative by applying psychoanalytic theory to Amharic novels, offering fresh perspectives on the emotional and psychological aspects of Ethiopian literature.

In conclusion, this study seeks to fill a critical, methodological, and theoretical gap in Ethiopian literary studies by exploring how mourning, melancholy, and ambivalence are represented in Amharic novels, particularly within the context of personal relationships. Through a psychoanalytic lens, this research aims to illuminate the complex emotional dynamics that shape Ethiopian literature, specifically analyzing ambivalence in selected Amharic novels. The main emphasis of this study was the analysis and interpretation of how ambivalences are represented in selected Amharic novels. In line with this, the following research questions are specifically formulated:

1. How are the complexities of ambivalence represented in the selected Amharic novels?

2. What are the emotional, psychological, and interpersonal consequences of ambivalence in the selected Amharic novels?

## 3. Research objectives

The main emphasis of this study was the analysis and interpretation of how ambivalence is represented in selected Amharic novels. In line with this, the following specific research objectives were formulated:

1. To explore the representation of the complexities of ambivalence in the selected Amharic novels.

2. To investigate the emotional, psychological, and interpersonal consequences of ambivalence in the selected Amharic novels.

## 4. Significance of the paper to PLOS ONE

This paper holds significant relevance to *PLOS ONE* as it investigates the intersection of psychoanalytic theory and literature, providing a fresh perspective on how emotional complexities and ambivalence manifest in literary works, particularly within the context of Amharic novels. The significance of this study to *PLOS ONE* can be articulated in the following ways:

1. **Multidisciplinary Approach**: *PLOS ONE* is known for its interdisciplinary approach to publishing research that transcends traditional disciplinary boundaries. This paper bridges the fields of literary analysis, psychoanalysis, and cultural studies, offering insights that contribute to both psychological theory and literary criticism. By examining how the psychoanalytic concept of ambivalence is represented in literary texts, this study provides a valuable contribution to the understanding of human emotions and relationships, offering an innovative perspective for both literary scholars and psychologists.

2. **Psychological Insights into Literature**: The paper provides a nuanced psychological reading of literary works, particularly focusing on psychoanalytic interpretations of grief, emotional conflict, and identity. This approach aligns with *PLOS ONE*'s interest in exploring the psychological and emotional states of individuals and how these states are reflected in societal contexts.

3. **Cultural and Societal Relevance**: The paper's focus on Amharic novels brings attention to the cultural and social context of Ethiopian literature, contributing to the diverse and global scope of *PLOS ONE*. By analyzing how these novels reflect universal themes of loss, love, and identity through a psychoanalytic lens, the paper adds to the growing body of research on the intersection of literature, culture, and psychology in non-Western contexts. This relevance to global literature aligns with *PLOS ONE*'s commitment to publishing research that represents diverse cultural and intellectual traditions.

4. **Contribution to Emotional and Psychological Studies**: The paper offers an important exploration of the psychological states of ambivalence, melancholia, and mourning, which are central to psychoanalytic theory. By examining how these emotions influence characters' behaviors and relationships, the paper contributes to our understanding of

emotional complexity and conflict, offering new insights into the nature of human experience. This focus on emotional dynamics can complement existing research in psychology, mental health, and emotional regulation, providing valuable resources for further studies in these fields.

5. **Potential for Broader Application**: Although rooted in literary analysis, the insights from this paper can be applied to broader fields such as mental health, narrative therapy, and cultural studies. The paper's findings on emotional conflicts and psychological paralysis could have implications for understanding human behavior in therapeutic settings, particularly in the ways individuals process grief and navigate conflicting emotions. This makes the study not only relevant to literary scholars but also to mental health professionals, therapists, and anyone interested in understanding the psychological underpinnings of human experience.

In conclusion, the paper's interdisciplinary nature, its contribution to psychoanalysis and literature, and its exploration of cultural themes make it a significant addition to *PLOS ONE*. The study provides valuable insights into the psychological dimensions of literature and the human condition, aligning with the journal's mission to promote knowledge across disciplines and to foster a deeper understanding of the complexities of human emotions and relationships.

## 5. Review of related literature

Literature has long been regarded as more than mere entertainment. It serves as a profound reflection of the human condition, offering insights into emotions, experiences, and societal dynamics. As Wellek and Warren [8] emphasize, critical approaches to literature provide a rich framework for understanding both universal and individual aspects of human existence. Novels are seen as vehicles for exploring the complexities of human relationships, emotions, and actions, as they often reflect real-world circumstances that shape human nature [16]. According to Ramanathan [17], literature is not only a medium for expressing attitudes and feelings but also a tool for understanding the stresses, anxieties, and contradictions embedded in the human experience. Novels, which portray society's everyday realities, offer invaluable insights into how individuals cope with emotions such as grief, loss, love, and conflict.

Cuddon [18] highlights that psychoanalytic criticism is primarily concerned with the emotional and mental lives of characters, often focusing on the intricacies of character development rather than plot or action. This approach is particularly useful in analyzing themes such as mourning, melancholy, and ambivalence, which are central to psychoanalytic theory. Freud [19] distinguished between mourning and melancholia, viewing mourning as a natural response to loss, while melancholia is understood as an extraordinary, pathological state characterized by unresolved grief and a deep, often debilitating sense of self-blame. Lacan [20] expanded upon Freud's ideas, conceptualizing mourning as a process that connects the individual to the symbolic order through rituals and ceremonies. For Lacan, the failure to move beyond this process results in melancholia, where the individual remains trapped in a state of emotional paralysis, unable to reintegrate into the symbolic and social order. The distinction between mourning and melancholia raises a significant problem: how does this unresolved grief or emotional paralysis manifest in literary characters? Specifically, how does it influence their ability to reconcile conflicting feelings or navigate interpersonal relationships? By focusing on ambivalence as a critical emotional state, psychoanalytic theory enables a deeper exploration of how characters in selected Amharic novels experience and express the tension between attachment and detachment, love and hate, and hope and despair. This tension often reflects a failure to resolve psychological conflicts, leading to ambivalence as a pervasive theme within the narrative structure.

The concept of melancholy, as discussed by Zimmerman [21], has evolved from being considered a universal human experience of suffering to being understood as a cultural construct. In ancient Greece, for example, melancholy was thought to be caused by an imbalance of bodily humors, particularly an excess of black bile. Over time, however, the understanding of melancholy shifted, and it came to symbolize deeper existential questions about human suffering, loss, and mortality. As a cultural construct, melancholy is often linked to profound emotional states such as disenchantment,

discouragement, and dejection—feelings that resonate across time and cultures as fundamental aspects of human experience.

In psychoanalytic theory, ambivalence is often viewed as an integral aspect of emotional life. Ambivalence refers to the coexistence of contradictory feelings, such as love and hate, towards the same object. Swales and Owens [22] describe ambivalence as a psychological state in which opposing emotions are simultaneously present, stemming from the same source but existing together in a conflicted state. This concept is particularly relevant in understanding the development of the psyche, as ambivalence often arises from the internal conflict between desire and rejection, love and hate. It plays a critical role in understanding the emotional dynamics of both individuals and relationships. In literature, ambivalence frequently manifests in character relationships, where characters' conflicting emotions toward each other or themselves create tension and complexity within the narrative.

Character interactions in Amharic literature are often characterized by ambivalence, as people struggle with emotions such as attachment and detachment, love and hatred, or hope and despair. These emotional dualities give characters' emotional lives a sense of instability and are frequently the result of unresolved internal conflicts or external cultural pressures. A more thorough examination of these conflicts is made possible by psychoanalytic theory, which also provides insights into how ambivalence reflects more general concerns of identity, self-perception, and personal development. This lens can be used to analyze how literary characters, especially those in Amharic novels, navigate complicated emotional landscapes that expose their weaknesses and wants as they struggle with their sense of self and place within the larger societal framework.

Additionally, the psychological concept of melancholia can be seen in the works of authors exploring themes of alienation, loss, and existential questioning. These literary works often investigate the psyche of characters who, unable to move beyond their grief or disillusionment, become trapped in cycles of self-reflection and despair. This psychological paralysis can create profound emotional depth in the narrative, as characters struggle to find meaning and purpose amidst the turmoil of their inner worlds. The role of melancholia, therefore, in literature is not just to depict suffering but to explore the psychological mechanisms that underlie human resilience and transformation.

## 6. Theoretical and analytical framework

With an emphasis on ambivalence as a crucial meta-psychological element, this study uses psychoanalytic theory to investigate the intricate emotional landscapes and underlying psychological processes in literature. This framework analyzes how the human psyche is portrayed in literature, namely in Sisay Nigusu's *Yäqənat Zar (Zar of Jealousy)* and Gebyehu Ayele's *Emba ɨnna Saq (Tears and Laughter)*, drawing on the seminal writings of Sigmund Freud, Jacques Lacan, Eugen Bleuler, and Melanie Klein.

### 6.1 Freud's psychoanalytic theory

Sigmund Freud's psychoanalytic theory, which encompasses the concepts of the id, ego, and superego, offers a foundational framework for analyzing the unconscious desires, conflicts, and motivations of literary characters. In particular, Freud's structural model of the psyche, comprised of the id (source of primal desires), ego (the reality-oriented mediator), and superego (the internalized moral compass), is essential for understanding the internal contradictions and ambivalence that shape a character's psychological depth. These conflicting internal forces are often manifested through characters' behaviors, motivations, and relationships within the narrative, as they navigate the tension between opposing desires and ideals [23].

Freud's exploration of ambivalence is especially evident in his theory of mourning and melancholia, outlined in *Mourning and Melancholia* (1917). Ambivalence, in this context, refers to the coexistence of conflicting emotions of love and hate, attachment and detachment toward an object or person, which complicates the grieving process. In Freud's framework, mourning involves the ego's attempt to reconcile the loss of an object with the reality of its absence, while

melancholia represents a pathological form of mourning where the lost object is internalized, leading to a profound sense of self-criticism and self-loss. This internal ambivalence between holding on to the lost object and letting it go creates psychological turmoil and confusion, reflected in the character's ambivalent feelings of sorrow and anger.

These ideas are especially pertinent when examining characters who battle internalized losses and unresolved grief in the chosen works. Their emotional states, which impact their behavior and relationships, show a continuous struggle between opposing emotions of love and resentment and attachment and rejection. The ambivalence of the characters is evident in their erratic emotional reactions, where their behaviors and desires appear to be at odds, causing internal conflict that frequently results in unhealthy relationships or unsolved psychological problems. Thus, Freud's theories—which emphasize ambivalence in melancholy and mourning—provide important insight into the intricate emotional landscapes and motivations of these characters, shedding light on how internalized conflicts mold their identities and actions.

## 6.2 Lacan's psychoanalytic theory

Jacques Lacan's extension of Freud's ideas introduces a more structuralist approach, emphasizing the role of language and social structures in shaping the unconscious. Lacan posits that the unconscious is structured like a language and that human desire is shaped by the symbolic realm, which includes the societal and linguistic systems that govern our interactions and desires. Lacan's theory of the *mirror stage,* where the child first identifies with their reflection and forms an ego, further emphasizes how identity and selfhood are rooted in external, symbolic recognition [24]. Lacan's differentiation between the *Symbolic*, the *Imaginary*, and the *Real* adds a profound dimension to understanding the psychological lives of literary characters. The *Symbolic* relates to societal laws and language, the *Imaginary* deals with personal identity and the formation of the ego, and the *Real* represents the unspeakable, unmanageable aspects of existence [24].

Through the lens of Lacanian theory, this research explores how the characters in the selected novels are shaped by societal influences and linguistic structures. It also investigates how their desires remain elusive and unattainable, leading to internal conflict and psychological tension. Lacan's concept of the "desire of the Other" illuminates the characters' motivations, often driven by the need for recognition and validation from others, thus contributing to their inner turmoil (Lacan, 1977).

## 6.3 Melanie Klein's object relations theory

Melanie Klein's object relations theory enriches this analysis by focusing on the early emotional bonds formed with caregivers and how these relationships shape the psyche throughout life. Klein emphasized the importance of early childhood experiences, particularly the relationship between the child and the mother, in the development of the self and its emotional responses [25]. Klein's work on the depressive position, in which the child begins to mourn the loss of the idealized mother and confronts both love and aggression, is especially relevant for analyzing themes of loss and ambivalence in the selected novels. These early stages of emotional development are marked by ambivalence, where the child experiences both love and aggression toward the mother, creating internal anxieties and conflicts that persist throughout life [25].

Klein's insights into object relations theory are key to understanding how characters in the novels process grief and unresolved emotional tensions. These dynamics, influenced by early relationships, shape their interactions with others and their emotional experiences. Furthermore, Klein's understanding of the depressive position helps explain how characters confront and internalize losses, particularly in the context of idealized or lost objects, contributing to their ongoing struggles with ambivalence and psychological distress.

## 6.4 Eugen Bleuler's theory of schizophrenia and ambivalence

Eugen Bleuler (1914), a Swiss psychiatrist, made significant contributions to the understanding of mental disorders, particularly schizophrenia, and the role of ambivalence in psychological functioning. While Bleuler is best known for coining the

term "schizophrenia," his work also expanded psychoanalytic thought by emphasizing the importance of ambivalence in human psychology and its manifestation in both normal and pathological states.

Bleuler's concept of ambivalence is foundational in understanding the complexity of human emotional life. He defined ambivalence as the simultaneous existence of contradictory emotions toward a person, object, or situation, which can create psychological tension and confusion. This ambivalence is particularly evident in Bleuler's exploration of schizophrenia, where the individual experiences a splitting of the self and a profound disconnection from reality. In such cases, ambivalence is not merely a psychological trait but a central dynamic that governs the individual's experience of the world.

In Bleuler's view, the fragmentation of the self in schizophrenia is rooted in the conflicting emotional states that disrupt the individual's ability to integrate their internal and external worlds. This fragmentation, marked by a breakdown in the cohesion of the ego, results in a distorted perception of reality and the inability to reconcile opposing feelings. In Bleuler's theory, ambivalence contributes to the disintegration of the self, as individuals are unable to bring together their contradictory impulses and experiences in a harmonious way.

The relevance of Bleuler's theory to literature, especially in the context of the selected novels, lies in its exploration of psychological fragmentation and the complex interplay between contradictory feelings. Ambivalence, as Bleuler conceptualized it, can be applied to characters in the novels who experience internal conflicts, torn between opposing desires or emotions. These characters may display behaviors that seem incoherent or contradictory, as they struggle with unresolved psychological tensions that mirror Bleuler's ideas about the fragmentation of the self.

A lens through which to view the emotional complexity of literary characters enmeshed in a cycle of internal contradictions is provided by Bleuler's conception of ambivalence as an emotional and cognitive conflict. Characters in literature can be viewed as navigating a shattered sense of identity, unable to reconcile their conflicting goals, anxieties, and motivations, much like Bleuler's patients struggled with split selves.

*The analytical framework of this study is crafted by integrating insights from prominent psychoanalysts and relevant literature. This includes the perspectives of psychoanalysts on the concept of ambivalence. The literature reviewed encompasses seminal works by Freud (1915, 1917, 1953, 1969), Lacan (1936, 1953, 1977), Klein (1940, 1950, 2005), and Bleuler (1914), among others.(Fig 10*

Through the application of Freud, Lacan, and Klein's psychoanalytic theories, this study offers a sophisticated examination of the unconscious dynamics and emotional landscapes found in the chosen novels. The framework highlights the importance of ambivalence, melancholy, and sorrow in comprehending the characters' psychological problems and motivations. From this psychoanalytic perspective, the novels are seen as more than mere surface-level narratives but as intricate psychological dramas that delve into the depths of the human condition.

Through the use of psychoanalytic literary critique, this study explores the psychological subject of ambivalence in the chosen books Yäqënat Zār by Sisay Nigusu and Emba ɨnna Saq by Gebyehu Ayele. The analytical framework integrates a number of psychoanalytic theories, such as Kristeva's theories on depression, melancholy, and mourning; Melanie Klein's object relations theory, which focuses on early emotional attachment and loss; and Lacan and Freud's theories on unconscious dynamics and ambivalence (Fig 1).

All of the meta-psychological factors are thoroughly examined, but ambivalence—which is examined through opposing emotions, inner conflicts, and the simultaneous sensation of love and hate—is given special attention. Psychoanalytic ideas like introjection, projection, and cultural ambivalence are all included in this theme. By examining these intricate emotional landscapes, the study seeks to reveal the books' underlying currents of internal conflict, longing, and loss in order to provide fresh perspectives on their emotional complexity and psychological depth (Fig 1).

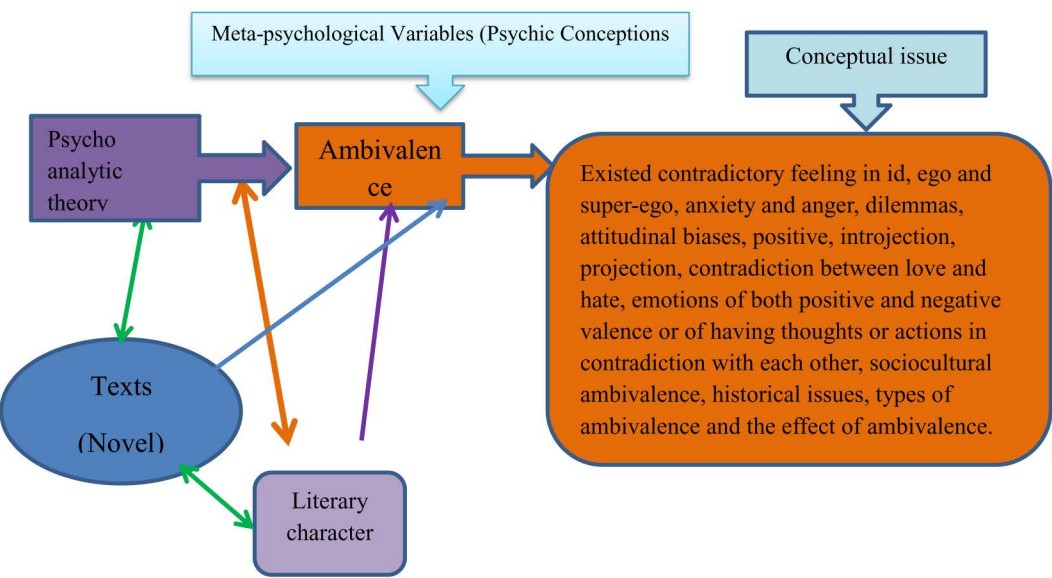

**Fig 1. Analytical framework.**

## 7. Methodology of the study

### 7.1 Research paradigm

In this study, the interpretivism research paradigm was employed. Interpretivism uses qualitative research methods that focus on individuals' beliefs, motivations, and reasoning over quantitative data to understand social interactions. It assumes that access to reality happens through social constructions such as language, consciousness, shared meanings, and instruments [26]. It integrates human interest into a study and assumes that access to reality/realities is only through social constructions such as language, consciousness, shared meanings, and instruments. Thus, to examine the portrayal of ambivalence in the selected Amharics, the interpretivism research paradigm was employed.

### 7.2 Research design and approach

This study employs a qualitative research approach, using a descriptive and analytical design. Descriptive research, as defined by Kothari [27], aims to present an accurate portrayal of the current state of affairs through surveys and fact-finding inquiries. In this case, the study focuses on an in-depth textual analysis of selected Amharic novels, guided by the principles of psychoanalytic criticism, to explore the themes of mourning, melancholy, and ambivalence.

The study is both theoretical and analytical, relying on close reading and interpretation to address the research objectives. Qualitative research, as Creswell (2009) notes, is essential for exploring and understanding the meanings individuals or groups attach to social phenomena. It enables a comprehensive examination of the research problem from multiple perspectives, facilitating a deep, interpretive analysis of the texts and the social issues they reflect. By using this approach, the study captures the complexity of the themes and contributes to theoretical development in the field.

### 7.3 Bases of text selection

The novels *Yäqënat Zär* [5] by Sisay Nigusu and *Emba ɨnna Saq* [6] by Gebyehu Ayele were selected for this study using purposive sampling, driven by the researcher's focus on mourning, melancholy, and ambivalence through a psychoanalytic lens. The selection is motivated by the absence of previous local studies that examine these themes within Ethiopian

literature using psychoanalytic theory, making these works a valuable contribution. The first criterion for selection is the subject matter, as both novels address themes of loss, grief, and internal conflict, offering rich material for exploring these emotions. *Yäqënat Zār* explores unfulfilled desires, jealousy, and the effects of loss, while *Emba ɨnna Saq* delves into trauma and the complexities of grief. These novels provide a foundation for understanding various forms of grief, including the loss of homeland, loved ones, and political ideals. The second criterion involves the text-to-theory approach, where the novels' elements of unconscious desires, repressed memories, and identity conflicts make them ideal for psychoanalytic interpretation, revealing deeper psychological dimensions. Third, both novels have garnered critical acclaim for their artistic achievements, using literary techniques such as symbolism, metaphor, and imagery to delve into the human emotional experience. Their intricate character development and plot progression contribute to their emotional depth. The fourth criterion is the representation of diverse perspectives on grief, allowing the researcher to explore how ambivalence is experienced differently by individuals and in varying social contexts. The fifth criterion is the diverse forms of grief presented, encompassing both pathological and non-pathological responses to loss and providing insights into the political, cultural, and artistic motivations behind the novels' portrayals of mourning and melancholy. Finally, the publication dates of the novels ([1] and [2]) offer a lens through which to explore changing representations of these themes in different historical and cultural contexts. Together, these novels exemplify the study's core themes and provide a comprehensive framework for analyzing the psychological dimensions of mourning, melancholy, and ambivalence in modern Amharic literature.

## 7.4 Procedures of data collection

The data collection process began with a close reading of the selected novels. The researcher carefully examined the original texts to fully understand their content, as these novels served as the primary data sources for the study. The next step involved selecting specific extracts, particularly those that highlighted themes of mourning, melancholy, and ambivalence. Alongside this, the researcher reviewed previous studies and consulted relevant library resources and journal articles to gain a deeper understanding of these psychological elements and gather supporting evidence for the discussion section. In the third step, the researcher thematized the identified data, categorizing it according to the key themes. The fourth step involved translating the selected Amharic extracts into English to ensure clarity and accessibility. Finally, the translated extracts were analyzed and interpreted using psychoanalytic literary theory, focusing on the three key psychic variables: mourning, melancholy, and ambivalence.

## 7.5 Data analysis

The analysis and interpretation of the selected Amharic novels were guided by the research questions and framed within psychoanalytic theory, with a particular focus on how mourning, melancholy, and ambivalence are represented and developed thematically in the narratives. The first step in the process involved identifying and thematizing relevant passages and scenes that depicted these psychological themes. These extracts were categorized into thematic clusters based on the specific emotions they conveyed. Next, each thematic cluster was analyzed using psychoanalytic concepts and theories, offering insights into the underlying psychological dynamics that shape the characters' experiences. Finally, the study assessed the manifestation of these emotions, identifying the dominant emotional themes in the texts. This step involved evaluating how the authors artistically express complex themes such as mourning, melancholy, and ambivalence, shedding light on their creative choices and the psychological depth embedded within the novels.

## 7.6 Translation approach

This study employed a literary translation approach to render the Amharic texts into English, aiming to preserve their artistic and aesthetic qualities while ensuring accessibility for readers in a different language [28]. The translation process integrated three main strategies: literal, cultural, and communicative approaches. The literal approach was used for excerpts where cultural or communicative context was less relevant, involving a direct word-for-word translation. However,

this method can sometimes obscure the original meaning or syntax [29]. The cultural approach, recognizing the importance of cultural context in translation, viewed translation as a distinct form of literature, considering cultural nuances and historical contexts [28] Finally, the communicative approach prioritized clarity and ease of understanding, making the text more accessible to the target audience [30]. These approaches were chosen to ensure linguistic accuracy, respect cultural influences, and facilitate effective communication of the intended meaning in the translated texts.

### 7.7 Credibility and trustworthiness of the research

To ensure the credibility and trustworthiness of this qualitative research, the researcher employed several validation strategies, as outlined by [12], which include credibility, dependability, confirmability, and transferability. Since this study is a textual analysis rather than a field-based study, participant validation strategies such as member checking and prolonged engagement were not applicable [31]. To enhance credibility, the researcher used methodological triangulation, aligning the research questions, qualitative design, and close reading techniques with the study's objectives. The study's transferability was ensured through thick descriptions that supported the findings and interpretations of mourning, melancholy, and ambivalence. Additionally, a theoretical framework, specifically psychoanalytic theory, was employed to guide the analysis and interpretation of the themes. Peer review also played a critical role in strengthening the study's trustworthiness; the research underwent external audits, and valuable feedback was received from seminars at Bahir Dar University, Debre Markos University, and national and international conferences. This process helped refine the analysis and ensure the credibility and confirmability of the findings [31].

### 7.8 Ethics statement

Because the study is based on textual analysis, no human subjects are used. The Bahir Dar University research rules, however, provided ethical confirmation for the study, and the university's examining board also assessed the project.

## 8. Results and discussions

This chapter presents the analysis and interpretation of two selected Amharic novels: *Yäqënat Zār (Zār of Jealousy)* by Sisay Nigusu [5] and *Emba ɨnna Saq (Tears and Laughter)* by Gebeyehu Ayele [6]. The primary focus of this chapter is to explore the ambivalent psyche of the characters, examining how psychological ambivalence is portrayed in the novels. The analysis will delve into the features of ambivalence, its underlying causes, and its effects on the characters' development and the narrative as a whole. Each novel will be analyzed separately under corresponding subsections.

### 8.1 The analysis of the character's ambivalent psyche in the selected novels

**8.1.1 Characters regret.** Ambivalence refers to the simultaneous experience of conflicting emotions or attitudes toward a particular person, object, or situation. In the context of anger and anxiety, ambivalence manifests as the coexistence of both emotions, creating a complex and often challenging emotional state. In the novels *Yäqënat Zār (Zār of Jealousy)* and *Emba ɨnna Saq (Tears and Laughter)*, the characters' ambivalence is closely tied to regret. This form of ambivalence involves conflicting feelings about past decisions, actions, or choices. Regret, a multifaceted emotion, arises when individuals believe they have made a mistake or wish they had chosen differently in a given situation. In *Yäqënat Zār*, the major characters, Semayneh and Hiwot, both experience regret as a result of their ambivalent emotions. This is evident in the following passage:

> "I am the guilty one; I don't blame anyone else. I neglected myself and others. I am unwell. It's best if I go to the clinic right away. My mind is in turmoil. I only got out of bed because I heard about Semayneh. Please call me or come over. "I have something to confess," she said with tears in her eyes. [261]

Based on this quote, Hiwot's aunt expresses deep regret not only for her husband's lies about her daughter, Hiwot, but also for her own actions. She admits to being emotionally disturbed, stating, "I was sick," which indicates her profound regret over the situation that unfolded. The ambivalence in her emotions is rooted in a complex interplay of regret, guilt, and suppressed feelings. As she confesses, "I am the one to blame; I don't judge anyone... I didn't care for myself or anyone else...," she acknowledges her role in the events that led to Hiwot's suffering, taking responsibility for her perceived failures. Her statement, "I am unwell... My mind is in turmoil," may conceal deeper feelings of anger and guilt. Rather than directly confronting her anger toward the situation or Semayneh, she seems to use her illness as a shield, distancing herself from her emotions and claiming a false sense of peace—possibly a defense mechanism to avoid dealing with the overwhelming guilt and anger she feels.

It is only when she hears about Semayneh's potential fate that an emotional release occurs. She seeks him out, perhaps to unburden herself of the secret she has been holding and to express her repressed anger. The anger she previously masked might resurface during this moment of confession, adding complexity to her emotional state.

After hearing the truth behind the story, Semayneh is also deeply upset and expresses his own regrets, stating:

> "What kind of deceitful woman she is!" At first, she lied to me, claiming to be my daughter when she wasn't even related to her. And now, she turns out to be the advisor of the sorcerer! No one has ever mocked me like her!" he exclaimed. "Calm down, my son," said the father. "What matters most is uncovering the truth. We should confront that witch. If what she told us is true, we will seek retribution." The mother became enraged. [32, p. 270]

The quote illustrates the deep regret and ambivalence experienced by Semayneh and his mother, emotions that are intricately tied to the human condition. Regret often arises from mistakes, missed opportunities, or loss, leading to feelings of sadness, guilt, shame, and anger. For Semayneh, his regret is rooted in the deception by W/ro Mulunesh, leaving him torn between conflicting emotions. This emotional turmoil reflects a profound internal conflict, encapsulating the complexity of ambivalence in his psyche.

When regret and ambivalence converge, they create a challenging emotional landscape. In Yäqënat Zār, Hiwot, the second major character, is also consumed by regret. She is constantly reflecting on her past choices and their devastating consequences, particularly the loss of her loved ones. Her guilt over the tragic events in her life, losing her family and her lover, intensifies her internal conflict. Hiwot's emotional turmoil is reflected in her sense of hopelessness and self-blame:

> "I don't see any hope. It is dark for me. It happened to me. I feel guilty... God has been so cruel to me! He took my mother and father away from me when I was very young. And now he took my lover. Why did he make me suffer like this?!" [32, p. 268]

Hiwot's regret, rooted in her tragic past, fuels her ambivalence, making her feel paralyzed and unable to move forward. Her internal conflict, driven by both guilt and resentment, traps her in a state of emotional and cognitive paralysis. This ambivalence leaves her unable to make decisions, rendering her stuck in a cycle of indecision and uncertainty. She longs to rewrite the narrative of her life but feels overwhelmed by the weight of her choices and the outcomes they've led to.

The emotional gravity of Hiwot's experience is encapsulated in her lament: "Now he has taken away my beloved lover. Why did he make me suffer like this?!" In this moment, regret and anger merge, reflecting the intensity of her ambivalence. These emotions often arise when a person is torn between conflicting desires or struggles to accept the outcomes of past decisions. Hiwot's regret is so overwhelming that it clouds her judgment and drives her to self-blame.

Similarly, Medemdya's conversation with Hiwot reveals a shared emotional state of regret and mourning. Both characters are haunted by the irreversible consequences of their past actions, and their shared sense of helplessness deepens their mutual ambivalence. This emotional bond further intensifies their inner struggles and adds complexity to their relationship.

In summary, Hiwot's ambivalence, shaped by regret and self-blame, manifests in emotional paralysis and an inability to move forward. The internal conflict she experiences underscores the profound impact of regret and its ability to entrap individuals in a cycle of uncertainty and inaction.

> She hid until she reached the place (America) where her lover was supposed to be. However, her efforts were in vain. Yesterday, she received a letter informing her that her boyfriend had taken his own life, leaving her bedridden from grief. Throughout the night until now, I've been calling her incessantly to find a house to rent and move her out of the hotel. Today, in the middle of the day, such a shocking event occurred,"he said with sadness evident in his voice. [32, p. 279]

The above statement suggests that Medemdmya is deeply irritated by the letter from Hiwot's boyfriend, Henok. Regret, when it takes hold, can undermine one's confidence in their decision-making abilities. This lack of confidence can manifest as ambivalence, making it difficult to make choices, leading to inaction or indecisiveness. Freud's theory on guilt provides further insight into this emotional turmoil. He posits that guilt stems from two sources: the fear of authority (the discovery and loss of love) and the fear of the superego. The fear of losing love carries moral implications, often manifesting as a "bad conscience." This form of guilt, which Freud refers to as "social anxiety," begins unconsciously and eventually evolves into conscious guilt. Regret, as a psychological trait, can therefore be seen as a key source of ambivalence, as individuals struggle to reconcile their actions with their desires and moral values. In line with this, Medemdmya's deep regret over the events in Hiwot's life is compounded by his earlier attempts to help her reunite with her lover in the United States. Despite his best efforts, the unfortunate circumstances Hiwot faces only intensify his feelings of regret. Ultimately, Medemdmya accompanies Hiwot to her aunt, a decision that reflects both his emotional burden and his desire to support her through her suffering, as conveyed in the following quote: "Don't forget me," she whispered. Huh-h "Remember me always. I am your mother. Forgive me for my mistakes..." Her voice trailed off weakly and softly, and then she fell silent. She drew her last breath in front of her daughter, bidding farewell to Hiwot. [32, p. 283]. This quotation presents Hiwot's mother's last words. She begged Hiwot to forgive her and not to forget her. In response to this, Hiwot said, "Mother... Mommy!... Mommy..." Despite Hiwot's desperate cries, there was no response. She had slipped into eternal slumber." [32]. p. After this, Hiwot falls into deep depression following her aunt's death. Her regret is rooted in the series of losses she has experienced throughout her life. Regret triggered by loss is a particularly powerful source of ambivalence, adding significant emotional complexity to characters and their internal struggles. When people experience loss, they often become fixated on what they've lost, replaying memories and longing for a past that can never be recaptured. This intense attachment to the past creates a sense of ambivalence towards the present and future, as individuals may feel torn between holding on to what was and facing the reality of what is. For Hiwot, this emotional conflict deepens her sense of regret, further complicating her ability to move forward.

In the novel *Emba ɨnna Saq*, the characters' ambivalent feelings stem from the regret they experience in their lives. Regret is a negative emotional response to a decision or action that one perceives as unfortunate or mistaken. It involves a deep sense of sorrow, remorse, or disappointment about past choices. Regret can arise from various sources, such as missed opportunities, unfulfilled goals, or the realization of the consequences of a particular decision. In Emba ɨnna Saq, the two main characters, Besrat and Sosina, are both burdened by regret in their lives. Their sense of regret creates emotional turmoil and ambivalence, as they struggle to reconcile their past decisions with the realities of their current situations. This internal conflict and sense of aggregation feeling overwhelmed by the weight of their past shapes their actions and emotional states, as illustrated in the following passage: … "He was anxious and broke down in tears. He longed to locate Sosina, witness her wounds, embrace her, and cry, releasing the pent-up emotions through tears." [12, p. 102].

In *Emba ɨnna Saq*, Besrat deeply regrets not being there for his wife while she is in the hospital. The provided quote reveals that Besrat is overwhelmed by a mix of emotions, including worry, regret, and anger. Specifically, the passage

describes how Besrat is anxious and bursts into tears, feeling a strong urge to find Sosina. He wants to check on her injury and express his emotions by holding her, crying, and releasing the pent-up feelings he's been suppressing. From this extract, it's clear that Besrat is grappling with ambivalence. He is torn between conflicting emotions, perhaps caught between his regret for not offering support and his anger, either at himself or the situation. His internal struggle creates a sense of emotional turmoil, which reflects the complexity of ambivalence. His worry and longing to comfort Sosina underscore the depth of his regret, making it a central driver of his emotional state. This passage exemplifies how regret can lead to a complex mix of emotions, leaving a character in a state of ambivalence, as shown in the following quote:

> "He lowered his voice as if he was talking to himself, saying, "Who does she have? She (her mother) is the only one in the world who has lost her. She told me that I was her reliance, but I didn't reach her." [12, 140]

The above statement indicates that Besrat was regretful due to the internal conflict in his mind. He was in guilt and self-blame. This shows that the person who didn't help may be wracked with guilt for their inaction, wondering if they could have made a difference. This can lead to a constant internal struggle between feeling responsible and helpless. Likewise, the second character having ambivalent feelings is Sosina "What can I do for you? I am a woman who can't do anything to eat; I will not harm my enemy that I will not use; as she spoke, she broke down in tears, and the black Sharp she had worn closed her eyes." [12, p. 139]. This indicates that Sosina is consumed by regret after losing her sight. She expresses feelings of helplessness, saying, "I was incapable of doing anything." Regret about being unable to do something she once could create a deep sense of ambivalence, layering her emotional struggle with frustration and confusion. This emotional conflict adds complexity to her internal turmoil as she contemplates her loss and the limitations it imposes on her. The phrase "As she spoke, she broke down in tears" signals an intense emotional breakdown, reflecting the depth of her inner conflict. This breakdown can be seen as a manifestation of her inner turmoil, likely driven by conflicting emotions like regret, guilt, frustration, and possibly a sense of failure. The weight of these emotions, compounded by the drastic change in her circumstances, leads to an overwhelming surge of feelings that she is unable to contain.

**8.1.2 Characters anxiety.** Freud argued that opposing forces coexist within every individual, and their conflict gives rise to ambivalence [19, p. 766]. In everyday life, this manifests in our relationships, where we might simultaneously experience love and attachment to someone while also harboring feelings of anger, resentment, or even a desire for separation. For instance, we may love our family deeply, finding joy in their presence, yet occasionally feel frustration, annoyance, or a need to escape the closeness. This internal push-and-pull is at the heart of ambivalence, as Freud suggests. In the novel *Yäqënat Zār*, the major character Hiwot is gripped by deep anxiety throughout the storyline. In the passage below, Hiwot is depicted in an anxious state, anticipating her lover's response, uncertain of what will happen. Her emotional turmoil continues until Medemdya arrives with a letter from Henok, which offers a potential resolution or at least an answer to the uncertainty that has been consuming her.

> When Medemdmya got up in the morning and went to the post office, his mind was full of different thoughts. He went to tell her the secret of Mr. Semayneh's illness and his admission to the hospital for psychiatric treatment that night. Hiwot was surprised by the situation and shook her head for a long time to express her sorrow, but she did not show any desire to meet with Heaven or to live back at her aunt's house. All her thoughts are only of Henok. [32, 263]

In the quoted passage, Hiwot's anxiety emerges from a complex interplay of factors: her concerns about her lover, Semayneh's illness, and various other stressors that exacerbate her emotional turmoil. She finds herself ruminating incessantly on her past interactions and decisions, replaying conversations and seeking hidden meanings, in an attempt to understand the source of her ambivalence. This ongoing mental churning manifests physically in sleeplessness, fatigue, and a lack of focus. Emotionally, Hiwot experiences frequent swings, moving erratically between hope and despair, anger and indifference, as she grapples with her uncertain situation.

Medemdya's news about Semayneh's hospitalization acts as a trigger, heightening her anxieties about her past with him, his condition, and the potential social repercussions. At the same time, she is confronted by unmet expectations, perhaps a hope to return home upon hearing of his breakdown, which leaves her feeling confused and disappointed. Her persistent focus on Henok, despite the turmoil surrounding Semayneh, signals an underlying anxiety about the stability of her relationship with him and the ambiguity of her future.

This emotional conflict, characterized by competing feelings of attachment and uncertainty, illustrates the complexity of Hiwot's inner turmoil. As she navigates these emotional entanglements, she experiences what might be described as "serial dissonance" [33, p. 175], shifting rapidly between conflicting emotions like anger, affection, and more anger, especially in relation to Semayneh. She also exhibits "layered conflict," a deeper, more intricate emotional state where multiple emotions intertwine, creating confusion and internal friction.

Hiwot's reluctance to meet Semayneh or return home reflects her ambivalence about confronting the past and the possibility of reconciliation. Her hesitance suggests a fear of revisiting old wounds or uncertainty about her commitment to Henok, indicating unresolved emotions or guilt that she has not fully addressed. Her emotional responses, both sorrowful and conflicted, emphasize her struggle with internal conflict and the uncertainty surrounding her future.

Pugmire's [33] theory on emotional depth and clarity further explains Hiwot's emotional state. He argues that profound emotional experiences require an internally coherent mind, free from conflict or ambivalence. In Hiwot's case, her emotional ambiguity, marked by, prevents her from experiencing the depth of emotions that might lead to meaningful resolutions or insights. The internal conflict she faces is symptomatic of an incoherent emotional state, preventing her from making decisive choices and leading to a state of confusion and stagnation.

In *Yäqënat Zār*, Semayneh also experiences intense emotional turmoil. His anxiety and anger, particularly concerning the disappearance of the witch, further emphasize the theme of ambivalence and emotional conflict. Both characters grapple with the challenge of reconciling their inner turmoil with the expectations and demands placed on them by their relationships and circumstances. This emotional tension underscores the broader themes of conflict, identity, and emotional clarity in the novel.

> Semayneh touched him since he missed the witch. He thought that surely there must be something hidden. He squinted his eyes and gnashed his teeth in frustration. "My tenant was a treacherous thief." If he were a real man, he wouldn't run away!" He glared at the old man. [32, 271]

The above quotation reveals that Semayneh's anxiety stems from the loss of the witch, an event that deeply unsettles him and fuels a sense of ambivalence in his life. His physical tension is evident in phrases such as "touched since he missed the witch" and "squinted his eyes and gnashed his teeth," which suggest discomfort, stress, and unease. Semayneh is far from relaxed; he is agitated by the situation. Internally, he is conflicted. Statements like "Surely there must be something hidden" and "treacherous thief" show his mind racing with suspicions and accusations as he struggles to make sense of what happened and find someone to blame. His frustration is palpable in the line, "If he were a real man, he wouldn't run away!" which expresses anger at the tenant's perceived cowardice and inability to take control of the situation. Semayneh's internal conflict is further highlighted by his ambivalence. On the one hand, he calls the tenant a "treacherous thief," yet on the other, he expresses a wish that the tenant had stayed, demonstrating indecisiveness and emotional turmoil. This internal tug-of-war between anger and a longing for clarity underscores his ambivalence. His accusations are not entirely certain; the phrase "missed the witch" hints at lingering doubt and a suspicion that there may be other, unexplored explanations for the disappearance. Additionally, his shifting focus glancing at the older man after accusing the tenant suggests a possible deflection of blame or an attempt to identify another source for his anxiety. Semayneh is not only angry and suspicious but also unsure, torn between conflicting emotions, and unable to settle on one explanation for the mystery that has consumed him. This inner turmoil and uncertainty reflect the complexity of his ambivalence. In

the novel *Emba ɨnna Saq (Tear and Laughter)*, the character's anxiety is a source of ambivalent behavior reflected in the novel, as the following quote indicates.

> Sosina's accident, Azeb's illness, W/ro Zewdie's condition, and the guard's anger and rage all made him shudder. The driver of the car pumped his fists and took a deep breath. ….. In this situation, he stopped and started the car. He was confused as to where he was going. The house to Zerihun to the hospital where Azeb is. [12, p. 103]

As the quotes indicate, Besrat's anxiety leads to ambivalent behavior. Phrases like "made him shudder" and "stopped and started the car" reveal a strong emotional response, showing how anxiety manifests physically through actions such as pumping fists, taking deep breaths, and shuddering. These physical reactions reflect his attempt to cope with internal turmoil. The confusion expressed in the quote, uncertainty about where to go or what to do, highlights the core of his ambivalence. The accumulation of stressors, including accidents, illness, and anger, intensifies his emotional conflict and compounds his inability to make clear decisions. This internal struggle is central to his ambivalence.

Similarly, Sosina in *Emba ɨnna Saq* experiences profound unease and apprehension, as seen in the following quote. This emotional turmoil, driven by various personal challenges, contributes to her own sense of ambivalence and internal conflict. Through these characters, Gebeyehu masterfully portrays the complex emotional struggles people face amid adversity. "During that time, she wiped her eyes; it is too bad, Tigest, Besrat was! By saying this, her tears stopped. "...I have a headache; please buy me an aspirin, Tigest" [12, p. 150]. The quote from Emba ɨnna Saq reveals Sosina's emotional turmoil, marked by anxiety and ambivalence. Her physical distress is evident when she wipes her eyes, suggesting recent tears and deep emotional pain. Verbally, Sosina's mention of another woman's sadness may reflect her own inner turmoil, projecting her feelings onto someone else. The repeated use of the word "sad" highlights the intensity of her emotions. Sosina's request for aspirin, typically used for headaches, symbolizes an attempt to address the physical manifestation of her emotional distress. This indirect request for help suggests a reluctance to openly express her sadness. Additionally, her contradictory plea for "patience" while also seeking a physical remedy illustrates her internal conflict wanting to express her feelings yet also striving to maintain composure. This passage underscores Sosina's struggle with emotional turmoil and ambivalence as she navigates both internal and external challenges.

## 8.2 The analysis of characters' ambivalence in psych models in the selected novels

### 8.2.1 The representation of ambivalence of characters id.
According to Sigmund Freud's psychoanalytic theory of personality, the id is the component of the unconscious mind driven by primal urges and desires, seeking immediate gratification of needs through the pleasure principle. In *Yäqënat Zār (Zar of Jealousy),* the major characters, Hiwot and Semayneh, each exhibit ambivalent feelings rooted in their unconscious desires. As illustrated in the following two quotations, Semayneh, one of the most jealous characters in the novel, experiences significant psychological ambivalence in his actions, torn between conflicting emotions and desires. His jealousy, driven by the id's demand for immediate satisfaction and control, plays a key role in the complex emotional dynamics of the story.

> Semayneh could not control his anger. The fact that he left the gun in the car made him even angrier. As he stood for a moment and snarled at the young men and shook his head, shaking his teeth, she grabbed his arm, fearing that he might get angry. [32,54]

The quote from *Yäqënat Zār* provides insight into Semayneh's internal conflict, highlighting the tension between his subconscious desires, driven by the id, and his conflicting actions. Semayneh's ambivalence stems from his unfulfilled desires, likely rooted in jealousy and possessiveness. His emotional turmoil is evident in his outbursts of anger ("to rush them and gnash his teeth") followed by restraint ("afraid of getting into a quarrel"), showcasing the internal struggle between aggression and

inhibition. Freudian theory suggests that the id, which seeks immediate gratification, drives Semayneh's intense possessive urges. His inability to control his anger and impulsive actions (e.g., leaving the gun in the car) reflects the id's dominance over his rational mind. The rapid shift from anger to fear further illustrates the tension between his id-driven aggression and the ego's concern for the consequences of his actions. This ambivalence reflects the conflict between unconscious desires and societal expectations, offering a deeper understanding of Semayneh's emotional volatility and character development. The quote below from *Yäqǝnat Zār* offers a fascinating glimpse into Hiwot's ambivalent behavior, showcasing how her id influences her emotional state and contributes to the complex dynamics between her and Semayneh.

> Hiwot was surprised at their condition and could not control her laughter. Even though she did not forget the anger of Semayneh, her laughter won her over and left. But before she could finish laughing, she felt Semayneh's punch on her jaw. The punch immediately knocked out her teeth, and she clutched the area with her hand and shook her head. [32, p. 59]

In *Yäqënat Zā*r, Semayneh's love for his wife is entangled with jealousy, leading to impulsive actions such as punching her. This can be attributed to the unchecked power of his id, driven by primal urges, as outlined in Freud's psychoanalytic theory. His ambivalence arises from the clash between love and jealousy while also displaying narcissistic tendencies. Similarly, Hiwot experiences ambivalence through internal conflicts and shifting loyalties. When she witnesses Semayneh's violent behavior but cannot suppress her laughter, it reveals a clash of emotions: her id's impulse to release tension through laughter conflicts with the superego's recognition of the danger. Hiwot's shifting loyalty, vacillating between acknowledging the seriousness of the situation and laughing at it, underscores her internal struggle. Both characters demonstrate the influence of the id. In Hiwot's case, her uncontrollable laughter signifies the dominance of the id, overriding rational thought. The abrupt shift from laughter to pain further highlights the id's volatile nature, where primal emotions can rapidly shift without regard for consequences.

In *Emba ɨnna Saq*, Besrat exemplifies ambivalence through conflicting emotions and desires, adding complexity to his character. His internal struggles create a multifaceted personality, shaped by the interplay of unconscious drives and rational thought.

> He returned and started the car and flew to Rasdesta Hospital. Along with the shock, Ato Gediwon used to choke him because the situation made him angry. He even forgot the car's lights, and it was only after traveling halfway that he remembered and turned them on. When he arrived at the hospital, the guards prevented him from entering. [12, p. 100]

In *Emba ɨnna Saq*, Besrat's actions and emotions illustrate ambivalence, driven by Freud's concept of the Id, primal, instinctual desires. Besrat's immediate response to the crisis, rushing to Rasdesta Hospital, reflects an impulsive reaction influenced by his Id. However, his anger and frustration with the situation manifest through physical symptoms, such as choking sensations, symbolizing an internal conflict between his desire to help and his emotional turmoil.

His forgetfulness about the car lights further underscores this ambivalence. Preoccupied with his emotions, Besrat's instinctual drive to act is distracted by anger, highlighting the tension between rational action and emotional impulses. The introduction of Ato Gediwon's anger intensifies this conflict, deepening Besrat's emotional struggle.

In short, Besrat's behavior, marked by impulsive action, anger, and forgetfulness, reflects the Id's influence, creating a complex emotional landscape. His conflicting instincts and emotions exemplify the ambivalence at the core of his character, as he navigates urgency, frustration, and internal strife.

In addition, the following two quotes also indicate that Besrat used his ID to express he was ambivalent:

> "Obey the law, or else you'll face the consequences." The words were accompanied by a threatening gesture, a boot aimed at his chin, as he stood waiting. [12, p. 103]

The guard's fury and rage surged towards him, startling him with a loud scream. He clenched his fists and took a deep breath as he punched the steering wheel. [12, p. 103]

Besrat's id is sourced from anger and forgetfulness in the above two quotes. Both quotes depict Besrat experiencing intense anger, adjusting his boot aggressively, and punching the steering wheel. This suggests anger might be a core aspect of his identity, potentially stemming from external factors like his job as a guard or internal struggles. Leaving the car lights off could be interpreted as a manifestation of unconscious emotions. The id operates on immediate needs and desires, sometimes overriding conscious thought.

The novel portrays a sense of ambivalence through the manifestation of Sosina's id, as exemplified in the following analysis.

"Release me! What do you intend to keep me here? You've isolated me from fellow humans, you heartless barbarians," she started sobbing and wailing. "Mother beasts, tyrants who neither fear humanity nor God, if conscience even exists for you! Oh, how can I escape this day of despair? No way out, as you keep circling around…" [12, p. 142]

The excerpt from *Emba ɨnna Saq* reveals Sosina's emotional turmoil, illustrating how her desires, driven by the Id, create ambivalence and desperate actions. Sosina's emotional conflict is clear in her fear, anger, despair, and longing for freedom, each emotion in tension with the others. Her emotional volatility portrays a character torn between opposing impulses, unable to reconcile her primal desires with the reality of her situation. Her contradictory actions—crying and pleading while cursing her captors—further highlight her ambivalence. This mix of vulnerability and aggression suggests an internal struggle between fear and a defiant urge to fight back, embodying her emotional conflict. Freud's id plays a central role in Sosina's ambivalence. The id, driven by survival, pleasure, and aggression, operates instinctively without regard for consequences. Sosina's captivity triggers her primal desire for survival and freedom, which clashes with her awareness of her powerlessness. This conflict between her Id's desires and the external reality fuels her ambivalence. In summary, Sosina's ambivalence stems from the clash between her instinctual desires for freedom and the harsh constraints of her captivity. This tension between primal urges and external realities shapes her emotional turmoil and drives her contradictory actions.

**8.2.2 The representation of ambivalence of characters' ego.** Ambivalence refers to the simultaneous presence of conflicting feelings or tendencies toward an object, creating emotional misery. Freud (1917) explains that ambivalence is a key factor in melancholia, which emerges after the loss of a loved object. In this state, the loved object is introjected, and the libido is withdrawn into the self, leading to identification with that object. The loss transforms into an ego loss, sparking a conflict between the ego and the superego. Similarly, in obsessional neurosis, ambivalence exists but remains directed toward the external object rather than the self. In the following quote, Semayneh's ambivalence reflects his struggle to satisfy his ego, as he is torn between conflicting emotions and desires related to his sense of self.

I have decided to marry you or kill you. Choose one of the two!" … Give me an answer; you have to choose between marrying and dying. [32], p.

In *Yäqĕnat Zār*, Semayneh's ambivalence reflects his internal conflict between control, conflicting desires, and self-importance, rooted in his inflated ego. Freud [34] suggests that love, initially narcissistic, originates from the ego's drive for self-pleasure and extends to objects incorporated into the ego. Semayneh's ultimatum, "Marry me or die," demonstrates his need for control and dominance, forcing Hiwot into an impossible position. This shows his desire for power, but also underlying ambivalence, as he likely recognizes the brutality of his demand. His inflated sense of self-importance distorts reality, as he reduces Hiwot to an object to fulfill his ego-driven needs, disregarding her autonomy and feelings.

Semayneh's selfishness and manipulation highlight his controlling nature, where his desires eclipse all other considerations, reflecting a clash between his ego-driven impulses and the consequences of his actions.

The following quote indicates Semayneh's ego as a source of ambivalence:

> Semayneh was stunned. She hadn't anticipated the pain would be so intense. Her expression didn't resemble her usual self; it seemed like someone else entirely. Alongside the physical pain, she trembled due to the circumstances of her life. "Don't cry; take it easy. It won't be a big deal. Since she angered me, you should give her a little slap," they advised him. [32,60]

Semayneh's behavior in the excerpt reflects ambivalence rooted in conflicts within his ego. This ambivalence arises from two main factors: internal conflict and conflicting needs. Initially, Semayneh shows empathy towards Hiwot, but this sentiment is quickly overshadowed by his willingness to consider violence, driven by external anger. This clash highlights his struggle between compassion and control. The second factor is the conflict between his id's desires and societal pressures from the superego. While Hiwot's vulnerability triggers empathy, societal expectations of masculine dominance push Semayneh towards aggression, creating a deep internal conflict.

Semayneh's ego plays a crucial role in his ambivalence, especially through reality testing and maintaining control. His ego recognizes the consequences of aggression, leading to hesitation, but also drives his need to assert dominance, possibly to preserve his image. This internal struggle between empathy and societal expectations manifests as ambivalence in his behavior.

Overall, Semayneh's ambivalence stems from the conflict between his id's aggression, his superego's moral constraints, and the desire for control. This dynamic reflects a character torn between opposing impulses—aggression and empathy driven by an inflated ego and a distorted sense of power. His behavior highlights the complexity of navigating internal forces and external expectations, making him a manipulative and controlling figure.

Freud [35] describes the ego as a source of anxiety due to its intricate and delicate relationship with the id and the superego. The id represents innate desires and needs, compelling it to seek satisfaction. Therefore, the id strives for the fulfillment of its desires and needs. The novel *Emba ɨnna Saq (Tear and Laughter)* portrays the protagonist's ego as originating from the experience of ambivalence.

> Go out; the scandal will follow if you refuse God's law. The stick eased him closer. He clenched his fist, poised it near his chin, and stood in anticipation. He thought of the chaos that would happen if he did that; he was afraid of the stick, so he swallowed his anger and quickly got out. [12, p. 103]

> When they were teasing him and patiently pushing him like a dog, he stood still and kept blinking his eyes like a smooth lie. The thin guard was surprised by his patience and opened the door... [12, p. 110]

The excerpts from *Emba ɨnna Saq* reveal Besrat's ambivalence, which is largely shaped by his ego's internal conflict and conflicting priorities. Besrat experiences a struggle between different desires and fears, which is evident in his hesitation. Initially, he may resist, but external pressures and the potential consequences of defiance ultimately push him to comply. This inner conflict reflects his ego's struggle between personal autonomy and external demands. Besrat's ambivalence is also driven by conflicting priorities. His ego grapples with the desire to avoid shame and personal scandal while simultaneously fearing physical punishment. This clash between self-preservation and the fear of social consequences causes Besrat to waver between compliance and resistance. The ego's role in Besrat's ambivalence is further highlighted by his awareness of external realities. Though he may harbor rebellious tendencies, his ego recognizes the risks of defying authority, including potential punishment. The fear of harm and the desire to maintain safety ultimately outweigh his urge to resist, leading him to submit.

In conclusion, Besrat's ambivalence is shaped by his ego, which mediates between competing desires, external pressures, and the fear of consequences. This internal conflict illustrates his reluctant compliance and sheds light on the power dynamics within the society depicted in the novel. Sosina, the second major character in the novel *Emba ɨnna Saq*, also experiences the manifestation of ambivalence within her ego. This is evident from the following quote:

"She said, "You are happy that I survived. "Yes, you are our important person, Sosina and the representative/substitute of our sister Kelmewa," said and gave a sign to them. "I don't think so, Ababa Gediwon," she said, leaning back. "... She kept fighting with strong emotion and looked up..." [12, p. 139]

The excerpt from Emba ɨnna Saq highlights Sosina's ambivalence, shaped by her ego's internal struggle with conflicting emotions and desires. Initially, Sosina feels relief at her survival, driven by her ego's instinct for self-preservation. However, this relief is quickly overshadowed by guilt and sorrow, particularly in reference to her deceased sister and her role as a replacement. This emotional conflict reveals the ambivalence within her ego as it navigates the tension between survival and grief. Sosina also experiences uncertainty and insecurity, as reflected in her questioning of her worth and future. The loss of her family and support system intensifies this insecurity, creating further internal conflict. This uncertainty, combined with societal expectations (such as being seen as a replacement for her sister), heightens Sosina's ambivalence. External judgment pressures her ego to balance survival with the need to conform to societal roles, despite the tragic circumstances. Ultimately, Sosina's ego is caught between self-preservation, grief, uncertainty, and societal pressures, resulting in a complex emotional landscape. Her ambivalence underscores the psychological challenges of survival and the difficulty of rebuilding life after loss.

### 8.2.3 The representation of ambivalence of characters' superego.

In Freud's theory, the superego represents the moral aspect of personality, embodying cultural norms, social values, and the internalization of parental guidance. It develops as a result of moral learning and functions as a conscience, directing individuals towards perfection. The superego is influenced by the external world and aims to regulate the desires of the id, balancing them with societal expectations. In *Yäqɘnat Zār*, both Hiwot and Semayneh exhibit ambivalence shaped by the influence of the superego. For Hiwot, this manifests in conflicting emotions as she navigates between the moral standards imposed by society and her desires. She experiences internal tension, caught between doing what is expected of her and her own emotional responses, revealing the pull of her superego alongside her more primal instincts. Semayneh's behavior is also influenced by the superego, particularly in his interactions with Hiwot. His sense of control and dominance can be seen as a way to uphold societal expectations of masculinity and authority. However, his actions reflect an internal conflict, as his id-driven impulses clash with the moral standards internalized through his superego. The tension between his desire for control and his recognition of moral limits creates an ambivalence that drives much of his behavior. In both characters, the superego plays a crucial role in shaping their moral struggles, balancing their personal desires with societal norms. The ambivalence they experience is a result of the conflict between these internalized moral standards and their more instinctual drives.

"Should I go and get a doctor?............" said After Semayneh called his mother outside, "Don't go away from her. Be careful," he warned them. Stay close to her and watch her movements. Don't stay away from her," and immediately pulled out the bedroom phone. [32], p.61

In *Yäqënat Zār*, Semayneh's actions reveal ambivalence shaped by the influence of his superego. His initial response, "Should I go and get a doctor?" demonstrates moral concern for others, reflecting the superego's guidance to act responsibly. His directive to call his mother and warn others to stay with the person in need further underscores his sense of duty and ethical obligation, in line with the superego's role as the moral compass. Freud's concept of the superego is evident

here, as Semayneh strives to fulfill his moral duties during a crisis. His decisive action of grabbing the phone to call for help reflects a sense of responsibility and adherence to ethical values.

However, Semayneh's ambivalence arises from the internal tension between his moral instincts and the uncertainties of the situation. While he is driven to ensure the well-being of others, the complexities of the moment introduce conflict, creating a sense of hesitation. This dynamic highlights the ambivalence between his moral responsibilities and the unpredictable challenges of real-life circumstances.

In a similar vein, *Emba ɨnna Saq* portrays Besrat's internal conflict, where his superego contributes to his ambivalence. In the first excerpt, Besrat's reluctance to interact with certain individuals suggests a moral disapproval, yet his compliance with external authority, *"hospital's permission,"* illustrates an internal struggle between personal values and societal expectations. This tension between moral judgment and external pressure fuels Besrat's ambivalence.

The second excerpt highlights Besrat's emotional turmoil, triggered by Zerihun's comment, *"My patience is over."* This remark prompts guilt and self-reproach, revealing the superego's role as an internal critic. Besrat's self-reflection and recognition of the ethical implications of his actions toward Sosina further emphasize the internal conflict between his desires and moral values.

These excerpts highlight three key aspects of Besrat's ambivalence: his hesitation and uncertainty in interactions, his emotional distress caused by the disapproval of his superego, and his difficulty in expressing his thoughts due to the internal struggle between personal desires and moral duty. The role of the superego is central in shaping this ambivalence, adding depth to Besrat's character and emotional complexity throughout the novel.

## 8.3 Types of ambivalence reflected in the selected novels

In *Yäqënat Zār* and *Emba ɨnna Saq*, ambivalence is a central theme explored through the major characters, reflecting Bleuler's (1914, cited in Koletzko, 2015) three manifestations of ambivalence: affective ambivalence, intellectual ambivalence, and ambitendence. Affective ambivalence is seen when characters like Hiwot and Sosina simultaneously experience opposing emotions, such as love and hate or relief and guilt, toward the same person or situation. Intellectual ambivalence appears when characters, such as Semayneh and Besrat, express contradictory thoughts or feelings, torn between conflicting beliefs or ideologies. Lastly, ambitendence is evident in the characters' conflicting desires, such as Hiwot's wish for freedom paired with fear of its consequences or Besrat's simultaneous desire to resist authority while wanting to avoid the risks involved. These varying forms of ambivalence create emotional and cognitive complexity in the characters, illustrating the internal conflicts they face as they navigate moral dilemmas, societal pressures, and personal desires.

In the novel *Yäqənat Zār*, the two major characters, Hiwot and Semayneh, reflect different ambivalent features as stated as follows:

Both of them got into the sheets together as if they were given different directions of emotions. "It is obvious that the daughter of a whore is a whore," said Semayneh, trying to catch his breath as he lay on the bed. [32.p.28]

Although he remembered this, he could not control his anger because the rumors he heard from Gorefu and the jealous zar made him drunk. As soon as Hiwot opened the door, trembling with fear, he jumped and shook her head. "Speak up, you whore!" Hyewot takes its breath so that it doesn't die without opening its eyes; it seduces it, and the sky is your soul. He let go, stroking her hair and holding her. [32], p.144

In *Yäqënat Zār*, the characters of Semayneh and Hiwot embody different forms of ambivalence that deepen the novel's exploration of emotional complexity. Semayneh's behavior toward his wife reflects affective ambivalence, where he loves her yet also engages in cruel acts like insulting and kicking her. His emotions are contradictory, reflecting a struggle between affection and aggression. On the other hand, Hiwot's ambivalence arises from a power dynamic rooted in fear and submission, further complicating their relationship.

Semayneh also displays intellectual ambivalence, particularly in a moment of self-awareness: "Because I am jealous of my love. I understand that I'm boring you" [32, p. 27]. This statement reveals his recognition of his jealousy and its

potential harm to his relationship, indicating intellectual self-reflection. However, his actions of kissing her neck and rubbing her hips while feeling jealous contradict his verbal acknowledgment of being boring, highlighting his internal conflict. This ambitendence (emotional and intellectual contradiction) underscores his struggle between emotional desire and intellectual understanding. Overall, the novel portrays a rich interplay of ambivalence, blending emotional volatility, intellectual contradiction, and internal conflict. These forms of ambivalence contribute to the depth of Semayneh and Hiwot's characters and their complex relationship.

In the novel *Emba ɨnna Saq*, the major characters reflect different features of both intellectual and ambivalent feelings, as supported by the following quote taken from the novel: Besrat was a major character who reflected ambivalent features.

"Ato Gediwon, keep running to get ahead of him without breaking even." But he was in trouble as he was getting lost, and things were getting complicated. He planned to destroy Besrat by Mulunesh and make her and her mother miserable (their slaves). His next trick was aimed at Sosina. But is it impossible? [12 p. 172]

The quote illustrates Besrat's internal conflict in multiple ways. First, he faces conflicting priorities, feeling the urgency to "get ahead" of Ato Gediwon, driven by ambition, yet he also grapples with uncertainty and fear. Besrat is distressed by Ato Gediwon's destructive plans but is unsure how to act, revealing an internal struggle between taking action and remaining passive. His fear is evident in the question, "What's going to happen?" expressing apprehension about the uncertain future and potential danger.

Besrat's intellectual and emotional ambivalence is also clear. He understands the gravity of the situation and the need for action, yet his moral concerns and emotional turmoil—especially empathy for Sosna—create hesitation. This tension between intellectual awareness and emotional conflict underscores Besrat's ambivalence, as he recognizes the urgency but is paralyzed by fear and moral uncertainty.

Sosina was a feeling, as reflected in the following quotes:

"... Why is it that you are talking to me separately? "Be calm; I'm telling you a beautiful secret, Sosina. Listen to me without regret so that it doesn't look evil," he said in a low voice. She gave him an ear that showed she was not interested in what he was talking about." [32, p. 173]

"It was signed secretly in your feeling (on your side)" without notifying others. "Ababa Gediwon is more of a mystery to me than to you." [12, p. 197]

Sosina's ambivalence is reflected in her interactions with Mr. Gediwon and other characters. She displays internal conflict when she hesitates to engage in a secret conversation, questioning, *"What is it that you are talking to me about separately?"* This reluctance suggests suspicion, loyalty to others, or fear of the potential consequences. She also exhibits disinterest, as seen in her response, *"Ababa Gediwon is more of a mystery to me than to you,"* indicating that her focus is more on Ababa Gediwon than the secret itself. Lastly, Sosina shows passive resistance to the power dynamics at play, as her reluctance and disengagement hint at subtle resistance to manipulation.

In her conversation with W/ro Zewdie, Sosina's ambivalence is further revealed. She expresses fear over the threat of death but also shows acceptance when she says, *"Let me die for your mother, as you have done for me."* This acceptance suggests resignation, as Sosina views death as a possible relief from her suffering. Her conflicting emotions of fear and resignation reflect the depth of her internal turmoil and the complex nature of her emotional state.

And after she was with Besrat and Besrat's sister Azeb, she was in ambivalent feelings, as supported by the following quotes:

If you think that way, you can take someone decent and make them do the kind of work your mother did."Hearing this, Sosina's patience snapped. Azeb's situation already weighed heavily on her. Tears welled up, threatening to spill.

Screaming felt tempting but ultimately unhelpful. Unable to bear it any longer, she desperately wanted to turn away. Rising abruptly, she began to pace slowly. [12, pp. 198–199]

The quoted passages illuminate the multifaceted nature of Sosina's ambivalence, encapsulating her feelings of frustration, submission, emotional conflict, and a desire for escape. Firstly, the tension between frustration and submission is palpable as Sosina grapples with Azeb's expectations, which clash with her own beliefs. While she harbors impatience and a yearning to voice her dissent, she hesitates to challenge the authority figure directly, hinting at a sense of submission or reluctance to confront. Secondly, her internal struggle is evident in the tears she attempts to suppress, reflecting the emotional turmoil stemming from conflicting desires and obligations. Lastly, her abrupt physical movement, pacing slowly, signifies a subconscious need for a temporary escape from the overwhelming emotions and complexities of the situation. Collectively, these layers of ambivalence paint a poignant portrait of Sosina's inner conflict, showcasing her attempts to navigate a challenging scenario fraught with uncertainty and conflicting emotions.

In the end, when Besrat comes to her, she is in complex feelings, as stated in the following quotes:

Are you still alive, my dear Besrat? Thank God for rescuing my soul from the grave,"she said, embracing him tightly and crying tears of joy. [12, p. 189]

"Oh, indeed, you are my precious one, and I am yours." "No, my beloved Besrat, I am your burden." "What a dilemma! In what language shall I explain to you?" [12], p.

"Everything will be revealed to her," she said, her eyes shining with inspiration. "Thank you, dear Besrat," she laughed, tears of happiness streaming down her cheeks as she hugged him tightly. [12], p.

In *Emba ɨnna Saq*, both Sosina and Besrat display ambivalence in their interactions, characterized by contradictory emotions and internal conflict. Sosina's self-doubt is evident when she questions Besrat about his happiness, showing uncertainty, yet she also seeks comfort by leaning on him. Besrat's response, while reassuring her of his love, reveals his own ambivalence. He expresses deep affection but also defensiveness, suggesting frustration or confusion about her doubts. Both characters embody contradictory desires and emotions. Besrat's ambivalence stems from a conflict between his ambitions for power, love, and control, which leads to morally ambiguous actions. Sosina's ambivalence revolves around her mistrust of Gediwon, torn between defiance and submission as she navigates a dangerous situation. This complexity in their emotions and desires deepens their character development and drives the narrative forward, adding richness to their dynamic.

### 8.4 The impact of characters' ambivalence in the selected novels

This analysis section focuses on two novels: *Yäqǝnat Zār* and *Emba ɨnna Saq*. It explores the impact of ambivalent experiences and feelings on the characters Semayneh and Hiwot in *Yäqǝnat Zār* and Sosina and Besrat in *Emba ɨnna Saq*. The analysis delves into the social, psychological, and cultural effects of these ambivalences on both the main characters and other major and minor characters within the novels.

**8.4.1 The impact of characters' ambivalence on him/herself.** Ambivalence (as in practical conflicts, moral dilemmas, conflicting beliefs, and mixed feelings) is a central phenomenon of human life. Yet ambivalence is incompatible with entrenched philosophical conceptions of personhood, judgment, and action and is denied or marginalized by thinkers of diverse concerns. Ambivalence is a feeling that creates a negative effect on the people who have ambivalent feelings; thus, in the novel *Yäqǝnat Zār*, the major character Semayneh is one of the most jealous people who creates contradictory feelings in his life, which makes him lose many things. The following extract elucidates this point.

From this moment, his mind becomes increasingly erratic. He might speak coherently at times, but his words are non-sensical at other times. Even his voice reflects his fluctuating state. It can become so weak and raspy that it's difficult to understand him, yet he might erupt in sudden shouts. Despite these outbursts, he can also be attentive and take his time to respond. His outward behavior and vocal tone clearly mirror his inner turmoil. Every time the loss of his life dawns on him, he equates it to the loss of his entire fortune, sending him into a rage. He attempts to regain control by searching for a glimmer of hope to rebuild his life, but these efforts are often futile. The root of his problem lies in his unwavering belief in the sorcerer's words, which paint a bleak picture of his future. [32, p. 50]

In *Yäqënat Zār*, Semayneh's ambivalence, deeply rooted in his loss of his wife and wealth, profoundly impacts his mental and emotional state. His dependence on Hiwot highlights his inability to function independently, and his ambivalent relationship with her inflicts more pain than a clearly negative one. Semayneh's internal conflict is evident in his erratic behavior, fluctuating speech, unpredictable actions, and emotional swings between anger and fleeting hope. These contradictions reflect his inner turmoil, exacerbated by the sorcerer's ominous prediction of a *"dark future,"* which fuels his anxiety and doubt.

Semayneh's ambivalence leads to self-destructive tendencies, hindering his ability to form stable relationships and chart a clear life path. It also results in missed opportunities, as his fixation on the past prevents personal growth. Finally, the enduring internal struggle contributes to mental and emotional deterioration, hinted at through themes of depression and anxiety. Overall, Semayneh's ambivalence shapes his life in significant, negative ways, blocking progress and impeding his potential for positive change.

The effects of Semayneh's ambivalence on his life and social arena or interactions are stated in the quote below:

The guards stationed outside the perimeter of the tower, poised to protect it, responded to the guard's signal by firing a shot into the air and shouting, "Stop!" However, Semayneh remained undeterred. Attempting to flee at high speed, he was intercepted by the police, who punctured the car's tires with gunfire. Still determined, Semayneh leapt out of the vehicle, only to be met with bullets from the police as he tried to escape. As the police approached him methodically, ensuring he remained calm and subdued, he succumbed peacefully, his voice silenced by the officers' bullets as he found his final resting place. [32, pp. 275–276]

Semayneh's ambivalence drives him into internal conflict, isolation, and desperation, ultimately leading to his loss of agency. The extract shows how he's torn between conflicting desires: his urge to escape versus a longing for peace or surrender. His refusal to stop, despite imminent danger, reveals his inner struggle and drives him to make self-destructive choices. His attempt to flee alone highlights his isolation, as he faces his turmoil without support, exacerbating his recklessness and further deepening his internal conflict.

Semayneh's reactive nature, being chased, shot, and subdued, underscores his loss of agency. He becomes a pawn in his own battle of conflicting desires, leading to his tragic demise. His actions also affect others, notably putting his brother Flood in danger and causing chaos among the police, showing how his unresolved ambivalence impacts those around him.

Ultimately, Semayneh's death symbolizes the destructive power of internal conflict and the dangers of failing to resolve it. The extract portrays him in a state of liminality, trapped between conflicting desires and societal norms, which leads to his isolation and downfall. In conclusion, Semayneh's ambivalence isolates him, fuels his self-destructive behavior, and culminates in his death, serving as a cautionary reminder of the need to address inner conflict before it harms oneself and others.

Hiwot finds herself entangled in ambivalent circumstances, leading to a profound sense of hopelessness, as depicted in the following excerpts from the novel *Yäqǝnat Zār*:

"Yes, Hiwot, he was driven by Satan and died," said Medemdeya, tearfully. "I can't believe it!... No matter how much I despise him, I wouldn't wish for his death," she cried, clutching her head. "Is it possible? What will happen now?" Bahru Tsimo tried to console her. "Is my aunt dead?... Please don't hide it from me, Medemedmya." "She's not dead. They're under the care of doctors. Maybe they'll survive when they hear your voice."

In the novel *Yäqënat Zār*, Hiwot's ambivalence is fueled by a mix of grief, anger, and hope following her husband's death and her aunt's accident. Initially, she mourns her husband's loss, feeling sorrow and rage, which reflects the complexity of her relationship with him. As news of her aunt's accident emerges, Hiwot experiences a clash of despair and hope, torn between mourning and clinging to the possibility of her aunt's survival. This emotional turmoil forces her to reassess her identity and future, leaving her in a liminal state between grief and hope. Her internal conflict creates a chaotic emotional landscape, which impacts those around her, especially Bahiru, who tries to comfort her. Ultimately, Hiwot's ambivalence hinders her ability to fully process her grief, leaving her in emotional uncertainty.

Similarly, in *Emba ɨnna Saq*, Sosina's ambivalence deeply affects her mental, emotional, and physical well-being. She oscillates between moments of joy and despair in her relationship with Besrat, creating anxiety, indecisiveness, and insecurity. Her internal conflict prevents her from forming stable connections, contributing to feelings of loneliness and chronic unhappiness. Sosina's unfulfilled desires, jealousy, and past grief further strain her relationship with Besrat and cause emotional distress. These unresolved conflicts also manifest physically, leading to neglect of self-care and psychosomatic symptoms, perpetuating a cycle of suffering. In sum, Sosina's ambivalence results in a damaging cycle that impacts her overall well-being, emphasizing the profound effects of inner conflict.

As stated in the quote below, Besrat's ambivalence in the novel *Emba ɨnna Saq* also affects himself:

With shock, Ato. Gediwon felt choked, his anger boiling within him. He even forgot to turn on the car's lights, only remembering halfway through the journey. Upon reaching the hospital, the guards barred his entry [12, p. 100]

He's in trouble, tears streaming down his face, shaking him. … Man, who are you to disregard the warning and come here? [12, p.110]

He shook his head, realizing that she had lost her only cherished possession in this world. He shook his head again, speaking softly to himself. [12, p.146]

In *Emba ɨnna Saq*, Besrat's ambivalence significantly impacts his psychological, emotional, and social well-being. Psychologically, his inner turmoil leads to confusion and agitation, as shown by his boiling anger and choking reaction to shocking news. This emotional conflict creates internal tension, undermining his mental stability. Emotionally, Besrat's ambivalence results in intense feelings of anger, sadness, frustration, and guilt, which manifest physically in his tears and shaking body. These fluctuations contribute to emotional exhaustion and mood swings, deepening his sense of unrest.

Socially, Besrat's ambivalence strains his relationships. His internal conflict impedes his ability to communicate or make decisions, as seen when the guards bar his entry to the hospital. His obsession with revenge and distrust further isolate him from friends and family, who may not support his actions or fear becoming entangled in his quest. This emotional isolation exacerbates his difficulties in forming meaningful connections, limiting his empathy and fostering strained interactions.

Ultimately, Besrat's ambivalence traps him in a vicious cycle of anger, grief, and isolation, which hinders his emotional healing and social reintegration. His journey in the novel becomes a struggle to reconcile his desire for justice with the need to rebuild his life, highlighting the profound consequences of unresolved internal conflict on his mental and social well-being.

**8.4.2 The impact of the main character's ambivalence on other characters in the novels.** A person's ambivalent feelings have positive as well as negative effects on the other person. In the novel *Yäqǝnat Zār (Zar of Jealousy),* the ambivalent behavior of Semayneh is affected by the feelings and everyday activities of his wife, Hiwot, as stated in the quote, "In her laughter, there was a sense of perplexity." [32, p. 4]. And in the quote:

> Slowly, she approached the mirror with two conflicting emotions. Would she be able to live, always restraining her laughter, concealing her smile, and honoring her husband's wish?... She attempted. [32, p. 50]

In *Yäqënat Zār*, Hiwot's emotional and psychological struggles are deeply influenced by Semayneh's ambivalence. His erratic behavior and jealousy create an emotionally charged and unstable environment for her. Hiwot's depression and self-doubt are evident, as she questions whether she made a mistake in marrying him. The tension in their relationship is compounded by Semayneh's jealousy, which even leads to physical consequences for Hiwot, such as losing her natural teeth, symbolizing the emotional toll his conflicting feelings take on her.

While moments of affection and laughter punctuate their relationship, these brief respites are overshadowed by Semayneh's emotional volatility, which keeps Hiwot on edge. His unpredictability creates emotional confusion for her, as she struggles to navigate his fluctuating moods. Despite these challenges, Hiwot becomes more self-aware and resilient over time, learning to adapt to Semayneh's behavior and cope with the emotional turbulence.

However, Semayneh's controlling behavior exacerbates Hiwot's emotional distress. His demands that she "save her teeth" and "hide her smile" reflect psychological manipulation, undermining her autonomy and emotional well-being. The expectation for Hiwot to conform to his desires, to "fulfill her husband's will," erodes her sense of self-expression and identity, leading her to suppress her own needs. This dynamic creates a cycle of emotional strain, where Hiwot's inner conflict and loss of self weigh heavily on her.

In sum, the ambivalence in their relationship generates a complex emotional landscape for Hiwot. While there are moments of connection and passion, they are overshadowed by confusion, emotional manipulation, and a loss of identity. Semayneh's ambivalence leaves Hiwot in a fragile, psychologically unstable state, caught between affection and distress, ultimately undermining her emotional and psychological well-being.

However, while Hiwot's physical wounds may have healed, the scars on her heart remain unhealed. The memory of being beaten with a cane during her early days of marriage still fills her with rage. She grapples with the uncertainty of living with a jealous husband and a family that blindly supports him. The replacement of one of her natural teeth with a gold brace, though mending her smile, only exacerbates her inner turmoil. [32,74]

In the above extract from *Yäqënat Zār*, Hiwot's emotional turmoil stems from her husband's ambivalence, which swings between love and hate, deeply affecting their relationship. The contrast between her physical healing (her teeth) and the psychological scars from abuse reflects the ongoing trauma she endures despite outward recovery. The mention of Hiwot's enduring anger from being beaten with a cane during her bridal days underscores the lasting impact of abuse, which fuels her resentment. Her uncertainty about her future, expressed through her question of how long she can endure living with a jealous husband and an unsupportive family, reveals a sense of entrapment and loss of control. The symbolism of the gold brace, replacing her lost tooth, highlights society's tendency to mask physical damage without addressing deeper emotional wounds. Lastly, Hiwot's isolation and betrayal by her family emphasize her lack of support and protection, intensifying her feelings of abandonment and vulnerability. This extract underscores the complex interplay of psychological trauma, family dynamics, and emotional isolation in Hiwot's life.

Moreover, in the early stages of Hiwot's and Semayneh's relationship, Hiwot experienced anxiety and fear due to her conflicting emotions. She was not in love with her husband, and this made her feel depressed. Hiwot struggled with these emotions, finding it challenging to navigate her feelings and the complexities of her situation, as stated in the following quote:

Hiwot, in her own way, left an unsettling feeling behind her. Instead of experiencing sexual arousal and relaxation, her body was throbbing with fear. "What will he say?... How will he feel?... Is his jealousy based on the future, or has the past ruined him?" She pondered these questions, trying to decipher his emotions. However, Semayneh failed to grasp her anxiety. He interpreted her remarks as affectionate and drew closer to her emotionally, his body heating up and his muscles tensing. They both ended up under the sheets together, seemingly pulled in different directions. [32, p. 28]

In the above extract, Hiwot and Semayneh are in a sexual mood, but at the same time, Hiwot is afraid due to the loss of her virginity and what her husband feels about it. As stated in the extract, "Both of them entered the bed together to make love as if they were given in different feelings." [32, p. 28]. This indicates that the two characters were in two opposite feelings. Additionally, Hiwot was impacted by Semayneh's conflicted behavior, as the paragraph below shows that she felt pressured to alter her behavior as a result. Ambivalence adds complexity and depth to characters and creates a captivating and unpredictable narrative. It compels readers to engage with the characters' internal struggles and anticipate the consequences of their conflicting emotions and desires.

Semayneh is lying unconscious in the psychiatric hospital. The painkillers and sleeping pills given by the doctors kept him in a deep sleep for long hours. His mother and sister were standing at the hospital's entrance, welcoming the visitors who came to inquire about his condition. Gorfu is sent to find a prescription medicine. As the news quickly spread, many people returned to inquire about Semayneh's state. [32, p. 258]

Semayneh's experience exemplifies liminality, a state of in-betweenness driven by ambivalence, reflected in his intense emotional swings. His inner conflict is so profound that it manifests in unconsciousness, symbolizing his escape from the clashing desires and emotions overwhelming him. While the source of these desires remains unclear, they likely stem from societal pressures, unfulfilled dreams, or internalized expectations, all of which pull him in conflicting directions, creating immense mental strain.

Semayneh's withdrawal into unconsciousness also symbolizes emotional isolation, signifying his inability to reconcile his emotions or communicate his inner turmoil. The ambiguity surrounding his condition, coupled with the influx of visitors, highlights the uncertainty of his future. His path is tied to resolving his internal conflict, yet the text leaves this unresolved, emphasizing his ongoing struggle.

Viewing Semayneh's situation through the lens of liminality reveals his struggle to navigate a psychological limbo, caught between competing desires. This ambivalence drives his emotional turmoil, impacting his mental well-being, relationships, and future trajectory, as he remains trapped in a perpetual state of conflict.

Semayneh's ambivalence also affects other people, such as W/ro. Sirash and Gorfu:

W/ro Sirash, amidst her shock and anguish, let out a piercing scream and instinctively shielded her face with the pillow, unable to bear witnessing the distressing sight before her. Her cry reverberated, echoing through the heavens. The very demons who had once cheered him on as he proudly wielded the firearm now commanded him to take action. He discharged the weapon twice. The sharp retorts of the gunshots, coupled with the sight of W/ro Sirash's blood seeping through the blankets, intensified his inner turmoil. Hastily, he shoved his companion aside and hastily approached their vehicle. Gorfu's emotions rendered them momentarily immobilized; Semayaneh stepped in, taking hold of the situation, and swiftly started the engine, driving them away from the scene. [32, p. 275]

Semayneh's ambivalence in this extract profoundly impacts those around him, particularly W/ro Sirash and his brother Gorfu. The triggering effect of W/ro Sirash's scream amplifies Semayneh's internal turmoil, which manifests outwardly through violence, deepening his anguish. This illustrates how one person's inner conflict can invoke trauma in others, as seen with W/ro Sirash, whose own emotional scars are awakened.

The *"devils"* in Semayneh's mind symbolize his conflicting desires and anxieties. W/ro Sirash's scream emboldens these inner demons, pushing him further into emotional chaos and escalating his internal battle. This eventually leads to a violent act: the murder of W/ro Sirash. The *"warm applause"* of the devils and their demand for violence represent the clash between compassion and rage, illustrating how unresolved conflict can erupt into external aggression.

Semayneh's actions have a devastating effect on his brother Gorfu, who is paralyzed by shock. The consequences of Semayneh's choices ripple through his relationships, symbolizing the destructive impact of his ambivalence. The scene underscores how unresolved inner conflict can shatter trust and create chaos.

Ultimately, Semayneh's ambivalence catalyzes tragedy, propelling him toward violence and inflicting immense pain on W/ro Sirash. The scene highlights the destructive consequences of unaddressed inner struggles, emphasizing the critical need to seek support and resolve internal turmoil before it spirals into darkness.

In the novel *Emba ɨnna Saq*, both Besrat and Sosina exhibit ambivalent features that influence their interactions with other characters. Sosina's ambivalence, in particular, has both positive and negative effects on her relationships with others. The following quote exemplifies how Sosina's ambivalent feelings influence her interactions:

"She wiped her eyes like that and said, 'It's sad.' Tigest, Besrat was... and tears stopped her when she said that."... Said, "I am in a headache; please buy me an aspirin." [12, p. 150]

When I cry because of my bad luck, I leave my sorrow behind. My luck has been spoiled since I was conceived. She covered her face with a pillow and started crying. [12, p, 206]

My dear, Besrat, Besrat, to confirm it was him, she examined his eyes, nose, and neck with her fingers and hugged him. She buried her face under his neck and began muttering in a muffled voice in an indistinct (unclear) language. [12], p. 188]

In *Emba ɨnna Saq*, both Besrat and Sosina display ambivalence, which profoundly affects their interactions with others. Sosina's ambivalence, in particular, produces a mix of positive and negative outcomes in her relationships.

In the first quote, Sosina expresses sadness and tears while speaking about Besrat, suggesting conflicting emotions toward him. Her request for aspirin hints at an attempt to use her emotions to manipulate him, indicating her complex emotional responses. In the second quote, she reflects on leaving behind sorrow by crying due to her bad luck, showing her tendency to suppress negative emotions while using crying as a means to cope. The act of covering her face with a pillow underscores her desire for privacy when processing her emotional pain. The third quote highlights Sosina's affectionate behavior toward Besrat, examining his features and embracing him tightly. Yet, the indistinct language she uses suggests her difficulty in expressing her true feelings, adding complexity to her emotional connection.

Overall, Sosina's ambivalence creates internal turmoil and emotional instability, leading to rapid emotional shifts and contradictory actions. This, in turn, complicates her relationships with others, as her unpredictable nature and fluctuating empathy can unsettle those around her. Her tendency to withdraw emotionally and conceal her turmoil only deepens the emotional distance, resulting in a lack of support in her relationships. Sosina's ambivalence, as portrayed in the quotes, demonstrates the complexity of her inner struggles and their effects on those she interacts with.

Similarly, Besrat, the novel's central figure, embodies a blend of traits that greatly affect the lives of others. His ambivalent nature gives rise to both positive and negative consequences, leaving a significant and lasting impact on those around him.

Sosina's accident, Azeb's illness, W/ro Zewdie's situation, and the guard's fury all sent shivers down his spine. The driver tightened his grip on the wheel and drew in a deep breath. … In this state of uncertainty, he started the engine. Should he go home, to Zerihun, or to the hospital where Azeb was?. [12, p. 103]

He returned and started the car, speeding towards Rasdesta Hospital. Along with his shock, Ato Gediwon was infuriated, choking with anger. He forgot to turn on the lights and only remembered halfway there. Upon arrival at the hospital, the guards barred his entry. [12, p. 100]

Besrat's ambivalence, as shown in the excerpts, has both negative and positive effects on those around him, creating a mixed impact. On the negative side, Besrat's conflicting emotions and indecisiveness often create difficulties for others. For instance, in the case of Sosina's accident, Azeb's illness, W/ro Zewdie's condition, and the guard's anger, Besrat becomes overwhelmed and unsure of how to act. This uncertainty frustrates and worries those relying on him for guidance and support. Similarly, when rushing to Rasdesta Hospital, Besrat's impulsive actions, such as forgetting to turn on the lights, cause disruptions and potential danger, further adding to the frustration of those around him. On the positive side, Besrat's emotional openness and vulnerability can evoke empathy from others. His genuine care and dedication to his loved ones, despite his internal conflict, may foster a sense of shared experience and create deeper connections. People who recognize his urgency to help can appreciate his intentions, strengthening the bonds within his relationships. Overall, Besrat's ambivalence, reflected in his indecisiveness, forgetfulness, and impulsivity, leads to worry and frustration for those around him. However, it also allows for moments of empathy and connection, creating a complex dynamic within his relationships.

## 9. Conclusion

This study analyzed the representation of ambivalence in the contemporary Amharic novels *Yäqĕnat Zār* [5] by Sisay Nigusu and *Emba ɨnna Saq* [6] by Gebyehu Ayele, focusing on how these emotions shape characters' psychological and emotional landscapes. The study addressed two key research questions: How are the complexities of ambivalence represented, and what are its emotional, psychological, and interpersonal consequences in these novels?

Ambivalence in the novels is portrayed through internal conflicts, contradictory emotions, and unresolved dilemmas. Characters such as Semayneh and Hiwot in *Yäqĕnat Zār* and Besrat and Sosina in *Emba ɨnna Saq* experience ambivalence due to their conflicting ambitions, desires, and internal struggles. For instance, Semayneh grapples with the tension between his personal desires and familial obligations, while Besrat contemplates his position within a society grappling with change. These emotional conflicts are consistently portrayed in the novels through key narrative moments that show characters' vacillation between desires for self-fulfillment and societal constraints. Specifics, such as Semayneh's hesitation during a critical family decision or Besrat's reflection on societal expectations, highlight the deep psychological dilemmas faced by these characters.

The analysis of these conflicts draws on Freud's model of the id, ego, and superego. For example, in *Yäqĕnat Zār*, Semayneh's psychological conflict between his ambitions (id) and his family's expectations (superego) mirrors Freud's theory of internal struggle and decision-making. This model is applied to the characters' decision-making processes, showcasing how they navigate the competing demands of personal goals and external realities. The influence of ambivalence on their relationships, particularly in moments of emotional attachment and fear of change, is evident in the tension between characters, such as Semayneh's evolving relationship with his father and Besrat's difficulty in making choices in his romantic relationships.

Furthermore, ambivalence is central to the emotional dynamics of both novels, with these personal emotional struggles also reflecting broader societal themes. In *Emba ɨnna Saq*, the cyclical nature of grief experienced by the characters mirrors Ethiopia's historical turbulence, particularly the collective experiences of loss and displacement caused by social upheaval. The recurring themes of mourning and melancholy throughout the novel are woven into the characters' psychological journeys, reinforcing the larger historical and social context. Similarly, *Yäqĕnat Zār* explores the tension between tradition and modernity, where characters' internal struggles are reflected against the backdrop of shifting familial and societal expectations. These societal tensions contribute to the characters' ambivalence, showing how deeply personal emotional conflicts are intertwined with larger socio-cultural shifts.

In conclusion, the emotional and psychological struggles explored in this study offer profound insights into the human condition. The characters' ambivalence underscores the emotional complexities of loss, attachment, and fear of change, demonstrating resilience in the face of existential uncertainty. By linking individual emotional turmoil with broader societal and historical contexts, such as Ethiopia's historical tension between tradition and modernity and collective experiences of loss, the novels provide a lens through which we can better understand both personal and collective human experiences in times of crisis and transformation.

## 10. Recommendations and implications

Based on the findings, several recommendations and implications arise. Amharic writers, including novelists, poets, and filmmakers, are encouraged to explore a broader range of emotions, such as ambivalence, melancholy, and grief, and to incorporate themes related to Ethiopia's historical events. By diversifying emotional portrayals and examining multiple perspectives, storytellers can create more compelling and resonant narratives.

For literature educators and students, future analyses could apply different theoretical frameworks to explore these themes further. Expanding research to include a larger corpus of Amharic literature from various periods and genres would offer a deeper understanding of how emotions like mourning and ambivalence are portrayed across the literary tradition.

Subsequent studies could also explore the impact of cultural, historical, and socioeconomic factors on the portrayal of these emotions in Amharic literature. Additionally, comparative studies between Amharic novels and works from other literary traditions could provide valuable insights into universal and culture-specific representations of emotional states, contributing to the field of comparative literature.

In conclusion, future research can further enrich the study of emotions in Amharic literature by integrating cultural, socioeconomic, and historical contexts, thus expanding our understanding of how these complex emotions are portrayed and perceived in Ethiopian society.

## Acknowledgments

With deep appreciation, I would like to thank all the people and institutions that helped make this study paper a reality. First and foremost, I want to express my gratitude to Dr. Tesfaye Dagnew, my principal, and Dr. Tesfamaryam Gebremsekel, my co-supervisor, for their essential support and advice during the research process. Our research's emphasis and direction were greatly influenced by their knowledge and perceptions. Additionally, I want to express my gratitude to Dr. Ayenew Guadu for his insightful comments, advice, criticism, and encouragement during the research process. I am also appreciative of Bahir Dar University's English Language and Literature Faculty of Humanities for giving us the tools and assistance we required to finish this project.

## Author contributions

**Conceptualization:** Dawit Dibekulu, Tesfaye Dagnew, Tesfamaryam G/Meskel.

**Data curation:** Dawit Dibekulu.

**Formal analysis:** Dawit Dibekulu.

**Funding acquisition:** Dawit Dibekulu.

**Investigation:** Dawit Dibekulu.

**Methodology:** Dawit Dibekulu.

**Project administration:** Dawit Dibekulu, Tesfaye Dagnew, Tesfamaryam G/Meskel.

**Resources:** Dawit Dibekulu, Tesfaye Dagnew, Tesfamaryam G/Meskel.

**Software:** Dawit Dibekulu.

**Supervision:** Dawit Dibekulu, Tesfaye Dagnew, Tesfamaryam G/Meskel.

**Validation:** Dawit Dibekulu, Tesfaye Dagnew, Tesfamaryam G/Meskel.

**Visualization:** Dawit Dibekulu, Tesfaye Dagnew, Tesfamaryam G/Meskel.

**Writing – original draft:** Dawit Dibekulu.

**Writing – review & editing:** Dawit Dibekulu, Tesfaye Dagnew, Tesfamaryam G/Meskel.

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
