## [Decision Letter · Decision Letter 0]

Dear Dr. Dibekulu,

Thank you for submitting your manuscript to PLOS ONE. After careful consideration, we feel that it has merit but does not fully meet PLOS ONE’s publication criteria as it currently stands. Therefore, we invite you to submit a revised version of the manuscript that addresses the points raised during the review process.

We look forward to receiving your revised manuscript.

Kind regards,

Chetan Sinha

Academic Editor

PLOS ONE

Journal Requirements:

2. Thank you for stating the following in your Competing Interests section: “18136”

3. We note that your Data Availability Statement is currently as follows: the data is available with in the article

4. Please ensure that you refer to Figure 1 in your text as, if accepted, production will need this reference to link the reader to the figure.

5. PLOS does not publish ‘About the Author’ sections as part of the manuscript file. Please remove this section from your manuscript and instead include it as a separate, Supporting Information file: https://journals.plos.org/plosone/s/supporting-information.

Reviewers' comments:

Reviewer's Responses to Questions

**Comments to the Author**

1. Is the manuscript technically sound, and do the data support the conclusions?

Reviewer #1: Partly

2. Has the statistical analysis been performed appropriately and rigorously?

Reviewer #1: N/A

3. Have the authors made all data underlying the findings in their manuscript fully available?

Reviewer #1: Yes

4. Is the manuscript presented in an intelligible fashion and written in standard English?

Reviewer #1: No

Reviewer #1: I must commend author(s) for their attempts to examine multiplicity of identities across Amharic literature. It is pertinent given the transdisciplinarity and cross-cultural replicability concerning underrepresented Amharans. The indigenous knowledge, somewhat analyzed in the ms, provides certain depth to what it means reading Amharic literary works. However, some discussions are not so straightforward:

1. I wish the introduction had problematized the issue right away as opposed to seeing rehashed emphases of literary studies. This could be done by elaborating upon the 'sanitized' representation of Amharic literature across the wide-ranging studies of literature. That is, author(s) might better start with 1) problematization of issue and description of terms as used in ms, 2) thesis statement, 3) objective of paper, 4) significance of paper to PLOS ONE, and 5) review of literature. At the moment, the introduction does not seem to engage readers, belaboring on what it means to prioritize literature, which is a well-known argument already.

2. Strange to say, having the words "duality" and "identity" in title, author(s) make little to no effort in teasing out these key words which could have been a fascinating investigation. At the moment, author(s) dwell, in so many places in introduction, the centrality of literature and psychoanalytical domains that intersect with Amharic literature.

3. Author(s) assume readers are well-aware of the intricate sociocultural, sociopolitical nuances of Amharans. While author(s) highlight the diverse tapestry of political, social, and cultural dimensions associated with Amharans, nowhere in introduction or review of literature section do author(s) 'inform' readers of specific sociocultural, sociopolitical 'confrontations' or problematics (was there indigenous community bloodshed? was there a largely, overlooked third gender? were there territorial disputes marginalizing the indigenous?), which could have been the key to unlocking the entire study. Readers ought to know the multiple ways in which Amharans as society play significant role to the publication of select texts.

4. It goes without saying that literary studies employ, more often than not, textual analysis. Author(s) could have elaborated at length the importance of close reading of excerpts, but at the moment, author(s) seem to be fixated with notion of qualitative or otherwise, which does not seem to provide key bearings to how the ms is supposed to be positioned. Is this a literary or social science ms? If this is a literary work, close reading approaches should have been the key from which author(s) draw upon to introduce readers. At the moment, the research design needs a lot of work, particularly from literary studies' standpoint.

5. Currently, figure 1 has a lot of awkward spellings that could tarnish the journal's reputation. Please fix this.

6. While author(s) provide a section on framework, it seems that author(s) are interested merely in name dropping, listing who's who in psychoanalysis and its intersection of this in literature. Author(s) could have been more specific in the lenses used and introduced to readers particular theoretical/analytical configurations that are recontextualized to the investigation across Amharic literature. At the moment, author(s) do not tease out this, and assume readers are well-attuned to the specific operationalization of framework employed in the article. This needs to be revised precisely.

7. The notion of translated texts is interesting, but nothing has been said concerning verification of translation. Who verified what? What is academic qualification of these inter-rater experts? How many hours of translation training were received by these translator-experts? This information might better be considered by author(s).

8. At the moment, analyses are not tied to the notion of "duality" and "identity," and the many ways in which these manifest in the grand scheme of psychoanalytical work involving said Amharic literature. Author(s) merely analyze without making scholarly and analytical links with the lenses as mentioned at the beginning of ms.

For these reasons, I recommended a major revision to the article. I wish author(s) the very best in revising said ms!

**Do you want your identity to be public for this peer review?** For information about this choice, including consent withdrawal, please see our Privacy Policy

Reviewer #1: No

---

## [Author Response · Author response to Decision Letter 1]

29 Jan 2025

I have addressed all the reviewers' comments and highlighted the revisions in red. Additionally, I have ensured that all changes align with the journal's template. I kindly request the reviewers to consider my article, which is a psychoanalytic study focusing on individual character psychoses rather than communal psychoses. I am ready for further review and will make any additional amendments as needed. Thank you for your valuable feedback.

---

## [Decision Letter · Decision Letter 1]

Dear Dr Dibekulu,

Thank you for submitting your manuscript to PLOS ONE. After careful consideration, we feel that it has merit but does not fully meet PLOS ONE’s publication criteria as it currently stands. Therefore, we invite you to submit a revised version of the manuscript that addresses the points raised during the review process.

We look forward to receiving your revised manuscript.

Kind regards,

Kamalakar Surineni, MD, MPH

Guest Editor

PLOS ONE

**Journal Requirements:**

**Additional Editor Comments:**

Thank you so much and very much appreciate for addressing the reviewers' concerns, pls address these minor issues before considering for publication.

Reviewers' comments:

Reviewer's Responses to Questions

**Comments to the Author**

Reviewer #1: All comments have been addressed

Reviewer #2: All comments have been addressed

2. Is the manuscript technically sound, and do the data support the conclusions?

Reviewer #1: Partly

Reviewer #2: Yes

3. Has the statistical analysis been performed appropriately and rigorously?

Reviewer #1: N/A

Reviewer #2: N/A

4. Have the authors made all data underlying the findings in their manuscript fully available?

Reviewer #1: Yes

Reviewer #2: (No Response)

5. Is the manuscript presented in an intelligible fashion and written in standard English?

Reviewer #1: No

Reviewer #2: Yes

**Reviewer #1:**  I commend author(s) for addressing the comments.

However, the ms suffers from major sentence-level issues. I recommend that the ms be substantially edited for readers' comprehension before it can proceed to production.

**Reviewer #2:**  The author has addressed most of the the reviewer’s comments effectively. The revised manuscript shows clear improvements in structure, theoretical grounding, and use of psychoanalytic analysis. The study now presents a strong and original exploration of ambivalence in Amharic literature. However, the manuscript still needs proofreading to fix grammatical and language issues, and minor restructuring to improve clarity and flow especially in the introduction and theoretical sections.

**Do you want your identity to be public for this peer review?** For information about this choice, including consent withdrawal, please see our Privacy Policy

Reviewer #1: No

Reviewer #2: No

---

## [Author Response · Author response to Decision Letter 2]

26 May 2025

Subject: Revised Submission – PONE-D-24-51424R1

Dear Reviewers,

Thank you for your valuable feedback on our manuscript, "Exploring Ambivalence: A Psychoanalytic Analysis of Emotional Complexity in Selected Amharic Novels." We have carefully addressed all comments, with revisions highlighted in the tracked-changes file. Key improvements include:

Language & Clarity: Enhanced readability and corrected grammatical issues.

Theoretical Rigor: Strengthened framing and methodological transparency.

Structural Flow: Improved organization, particularly in the introduction and discussion.

Attached are:

Response to Reviewers (detailed point-by-point replies).

Revised Manuscript with Track Changes.

Clean Manuscript Version.

We appreciate your time and hope the revisions meet your expectations. Please let us know if further adjustments are needed.

Best regards,

---

## [Editor Report · Decision Letter 2]

Exploring Ambivalence: A Psychoanalytic Analysis of Emotional Complexity in Selected Amharic Novels

PONE-D-24-51424R2

Dear Dr.Dawit Dibekulu,

We’re pleased to inform you that your manuscript has been judged scientifically suitable for publication and will be formally accepted for publication once it meets all outstanding technical requirements.

Kind regards,

Kamalakar Surineni, MD, MPH

Guest Editor

PLOS ONE

---

## [Editor Report · Acceptance letter]

PONE-D-24-51424R2

PLOS ONE

Dear Dr. Dibekulu,

I'm pleased to inform you that your manuscript has been deemed suitable for publication in PLOS ONE. Congratulations! Your manuscript is now being handed over to our production team.

Kind regards,

on behalf of

Dr. Kamalakar Surineni

Guest Editor

PLOS ONE